# No carbon storage in growth-limited trees in a semi-arid woodland

R. Alexander Thompson [1] ✉, Henry D. Adams[1], David D. Breshears[2], Adam D. Collins[3], L. Turin Dickman [3], Charlotte Grossiord[4,5], Àngela Manrique-Alba [6], Drew M. Peltier [7], Michael G. Ryan [8,9], Amy M. Trowbridge[10] & Nate G. McDowell[11,12]

Plant survival depends on a balance between carbon supply and demand. When carbon supply becomes limited, plants buffer demand by using stored carbohydrates (sugar and starch). During drought, NSCs (non-structural carbohydrates) may accumulate if growth stops before photosynthesis. This expectation is pervasive, yet few studies have combined simultaneous measurements of drought, photosynthesis, growth, and carbon storage to test this. Using a field experiment with mature trees in a semi-arid woodland, we show that growth and photosynthesis slow in parallel as $\psi_{pd}$ declines, preventing carbon storage in two species of conifer (*J. monosperma* and *P. edulis*). During experimental drought, growth and photosynthesis were frequently co-limited. Our results point to an alternative perspective on how plants use carbon that views growth and photosynthesis as independent processes both regulated by water availability.

Drought constrains plant growth and photosynthesis by reducing cell turgor, preventing cell expansion and closing stomata, though these symptoms are often not simultaneous[1,2]. Growth tends to respond more rapidly than photosynthesis to water stress[1–3]. When drought inhibits growth[1] (herein *sink limitation*) ongoing photosynthesis can provide carbohydrates to be stored as non-structural carbohydrates (NSCs)[1,4,5]. NSCs, which include sugar and starch, provide an important source of metabolic substrate for plants during drought, after photosynthesis stops (herein *source limitation*)[1]. NSCs can be used to support respiration[6], synthesize defense compounds[7–9], avoid turgor loss[10], and delay plant death[11–13]. Ultimately, carbohydrate accumulation in plants experiencing drought-induced growth limitation remains an important yet unproven hypothesis in plant ecology. We test this core idea in plant drought response using mature trees with simultaneous measurements of photosynthesis, growth, NSCs, and plant water potential.

Drought-induced limitation to growth occurs as $\psi_{pd}$ becomes more negative, leading to a decline in cell turgor, preventing cell division and elongation[14]. Eventually, drought stress leads to stomatal closure, photosynthesis stops, and plants enter a state where survival depends on the consumption of stored NSC[1,11]. Photosynthesis may decline less quickly than growth during drought, but this has only been tested in a few species[2,3]. This tendency to overgeneralize results from a few organisms, often growing under experimental greenhouse conditions, and at early ontogenetic stages, to all plants may be the source of significant error in modern vegetation models[15,16].

Few studies use mature trees in field experiments to evaluate carbon source-sink dynamics under drought[15]. Surprisingly, no studies

[1]School of the Environment, Washington State University, Pullman, WA 99164, USA. [2]School of Natural Resources and the Environment, University of Arizona, Tucson, AZ 85719, USA. [3]Los Alamos National Laboratory, Earth & Environmental Sciences Division, Los Alamos, NM, USA. [4]Plant Ecology Research Laboratory PERL, School of Architecture, Civil and Environmental Engineering, EPFL, CH-1015 Lausanne, Switzerland. [5]Community Ecology Unit, Swiss Federal Institute for Forest, Snow and Landscape WSL, CH-1015 Lausanne, Switzerland. [6]Estación Experimental Aula Dei (EEAD-CSIC), Zaragoza, Spain. [7]Center for Ecosystem Science and Society, Northern Arizona University, Flagstaff, AZ 86011, USA. [8]Department of Ecosystem Science and Sustainability, Colorado State University, Fort Collins, CO 80523, USA. [9]USDA Forest Service, Rocky Mountain Research Station, Fort Collins, CO 80526, USA. [10]Department of Entomology, University of Wisconsin, Madison, WI 53706, USA. [11]Atmospheric Sciences and Global Change Division, Pacific Northwest National Lab, PO Box 999 Richland, WA 99352, USA. [12]School of Biological Sciences, Washington State University, PO Box 644236 Pullman, WA 99164-4236, USA. ✉e-mail: robert.a.thompson@wsu.edu

have simultaneously measured changes in growth, photosynthesis and starch accumulation along a $\psi_{pd}$ gradient on mature trees in the field[9]. To address these gaps, we leverage data from a 25-year, monthly $\psi_{pd}$ monitoring project (Mesita Del Buey, hereafter MDB) and a field drought experiment (the Los Alamos Survival Mortality Experiment, hereafter SUMO). MDB was established to monitor the long-term water status of mature trees in a piñon–juniper woodland of northern New Mexico from 1992 to 2016 and may be the longest continuous measurement of plant water potential[17,18]. SUMO was a multi-year field experiment (2012–2016) designed to manipulate temperature (+4.8 °C) and precipitation (−45%) consistent with climate change predictions. This integration of a long-term dataset with a neighboring manipulative study is ideal for evaluating the effects of water limitation on plants in the field.

Here, we ask two questions concerning plant carbon dynamics during drought: (1) Does growth stop before photosynthesis in response to declining $\psi_{pd}$? and (2) Does carbon storage increase as growth declines with more negative $\psi_{pd}$? Monthly measurements of $\psi_{pd}$, photosynthesis, and stem radial growth from the SUMO experiment were used to identify thresholds at which growth and photosynthesis reached zero (sink and source limitation, respectively), in two tree species that differ in tolerance to drought, *Juniperus monosperma* and *Pinus edulis* (Fig. 1). Thresholds for the cessation of growth and photosynthesis were defined as the *x*-intercepts of the species-level regressions of percent of maximum growth or photosynthesis against $\psi_{pd}$ (Fig. 1). We used these thresholds for the SUMO trees to evaluate the hypotheses that growth would stop before photosynthesis, driving an increase of starch in these trees.

## Results and discussion

Counter to our expectation, growth did not stop before photosynthesis (Fig. 1). Photosynthesis initially declined significantly faster (slope = −33.3, $P < 0.05$) than growth (slope = −26.8) for *J. monosperma* (Fig. 1B). Near a $\psi_{pd}$ of −3.5 MPa, both variables converged until crossing the x-axis at −6.86 MPa and −6.75 MPa for growth and photosynthesis, respectively (Fig. 1B, not significant). *P. edulis* also displayed strong patterns of co-limitation of growth and photosynthesis across the entire range of observed $\psi_{pd}$ (Fig. 1C; $P = 0.064$).

The distinct co-limitation patterns of each species reflect their differing tolerances to drought stress. *P. edulis* is considered isohydric compared to the relatively anisohydric *J. monosperma*[19]. Though many factors influence a species' position on the an/isohydry spectrum, two dominating factors are the regulation of leaf water loss[20] and hydraulic vulnerability[19]. The relatively early stomatal closure of *P. edulis* is an embolism-avoidance mechanism, preventing water loss through transpiration, avoiding hydraulic failure. *J. monosperma*, owing to its less vulnerable xylem, leverages osmotic adjustment to keep stomata open and potentially maintain growth under progressively lower $\Psi_{pd}$. While an/isohydry may explain the quantitative differences we observed (Fig. 1), we do not test this further. Instead, we focus on questions relating to the qualitative dynamics (i.e., we look within species or along common axes of variation).

*Can the co-limitation of growth and photosynthesis under progressively lower $\psi_{pd}$ be explained as the control of growth by photosynthesis (or vice versa) in these two species?* Prior work[21] has shown molecular evidence of coordinated growth and photosynthesis under low resource availability however, during drought, water availability is the primary factor limiting growth and photosynthesis. The co-limitation of growth and photosynthesis may be the result of a coordinated yet independent response to drought (Fig. 1B, C). In support of this view, we provide several lines of evidence. First, growth and photosynthesis showed no correlation for *P. edulis* only (Fig. 2A). Growth and photosynthesis were significantly correlated for *J. monosperma* yet very little of the variance could be explained ($R^2 = 0.05$). This suggests a latent, underlying driver of this variation. Trees of both

species could maintain relatively high growth rates under low photosynthesis (and vice versa). Second, growth and photosynthesis were coordinated only when water availability was low (perhaps driving the correlation in *J. monosperma*) during the summer drought and followed separate trajectories after the drought was relieved (Fig. 2B).

Growth and photosynthesis were qualitatively similar in both species during the 2013 and 2014 growing seasons (Fig. 2B). Both species experienced their lowest levels of growth and photosynthesis during the earliest parts of the growing season, consistent with low soil water availability. Following the late summer monsoon (July/August), photosynthesis exceeded growth in both species (~50% difference in both species; Fig. 2B). The absence of a proportional increase in growth rate with increasing photosynthetic rate implies growth was not limited by carbon availability.

In 2013, $\psi_{pd}$ averaged −2.91 MPa (±0.1) and −1.99 MPa (±0.04) for *J. monosperma* and *P. edulis*, respectively at SUMO. $\psi_{pd}$ was less negative in 2014 and averaged −1.74 MPa (±0.05) for *J. monosperma* and −1.53 MPa (±0.03) for *P edulis* at SUMO. 2013 at SUMO was the only year that *J. monosperma* ever experienced $\psi_{pd}$ low enough for growth and photosynthesis to cease. Although *P. edulis* fell below this threshold several times throughout the SUMO experiment, most of these observations were during the 2013 growing season (Supplementary Fig. 11; lines are estimates of growth and photosynthesis extrapolated from $\psi_{pd}$). This difference in $\psi_{pd}$ among years may explain the extremely low values of growth and photosynthesis in 2013, relative to 2014 (Fig. 2B). The growth of *J. monosperma* tended to track photosynthesis in 2014, like the patterns observed in 2013 though average rates were much higher in the early season of 2014. The growth rate of *P. edulis* also qualitatively tracked photosynthesis in 2014 but was sustained at a rate roughly 10% greater than the rate of photosynthesis. This suggests that even when photosynthesis is less than the demand by growth, the pool of available carbohydrates within a tree, both stored and incoming from photosynthesis, is sufficient to supply growth.

On average, NSC in the leaves and twigs of both species decreased with decreasing $\psi_{pd}$ (Fig. 3; slope = −1.2 for *J. monosperma*, −9.4 for *P. edulis*). Similar trends were observed for the bole and roots of *P. edulis*. In contrast, overall NSC in the roots of *J. monosperma* had little change with water stress (slope = 0.013, $P = 0.3$.). However, roots saw a relatively strong increase in sugar concentrations (slope = 0.1, $P < 0.05$) and decline in starch (slope = −0.05, $P < 0.05$; Fig. 3C). To support increased water acquisition from the soil and maintenance of cell turgor in the canopy under low $\psi_{pd}$ our data support starch conversion to sugar for osmotic purposes in this species[20] (Fig. 3C). Since *J. monosperma* maintained photosynthesis under relatively low $\psi_{pd}$ (Fig. 1), it is possible that sugars were transported from the canopy to roots, yet the slight trend in total root NSC does not strongly support this (Fig. 3C). *P. edulis* also exhibited steep increases in bole and root sugar concentrations (slope = 0.19 for the bole and 0.2 for the roots, $P < 0.05$; Fig. 3C) but virtually no change in leaf and twig sugar (slope = −0.007 for leaves, 0.005 for twigs, n.s.). The early onset of source limitation in *P. edulis* (Fig. 1) is known to preserve water storage in canopy tissues, perhaps avoiding the need for starch-to-sugar conversion in these tissues for osmotic purposes[20]. Although *J. monosperma* and *P. edulis* employ distinct physiological responses to water stress, our results indicate a decrease rather than an increase in overall NSC during drought (Fig. 3A, B).

We observed no significant change in NSC or starch with decreasing growth in *J. monosperma* (Fig. 4). Under extremely low growth, canopy sugar concentrations in *J. monosperma* increased logarithmically (slope = 1.14, $P < 0.05$). Mirroring the response to $\psi_{pd}$, *P. edulis* showed a significant decrease in leaf (slope = −0.54, p < 0.05) and twig (slope = −1, $P < 0.05$) starch which may have driven the significant increase in bole sugar concentrations (slope = 0.57, $P < 0.05$) as growth declined. Alternatively, leaf and twig starch in *P. edulis* may have been converted into sugar and used for metabolism, preventing

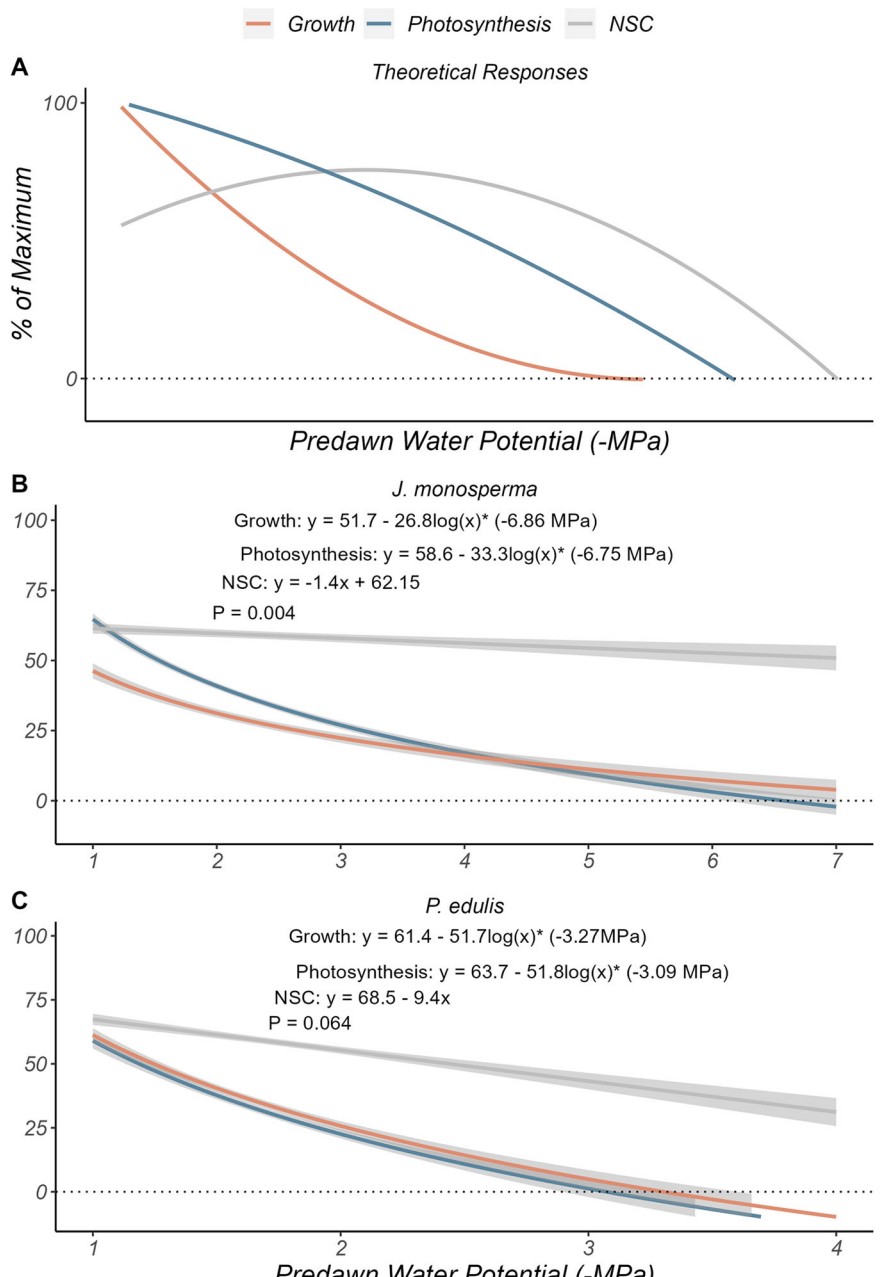

**Fig. 1 | No carbon storage for growth-limited trees.** Growth and photosynthesis are co-limited as $\psi_{pd}$ declines (**B** () and **C**). The expected increase in NSC accumulation as growth becomes constrained by drought is subverted given the rapid decline in photosynthesis. **A** The theoretical response of growth, photosynthesis, and NSC is shown based on predictions from theory[1]. **B**, **C** The observed response of growth, photosynthesis, and canopy NSC to drought suggests this theory may not apply to mature trees living in semi-arid environments. Lines in (**A**) reflect a loess smooth function. **B**, **C** Lines reflect point estimates of the regression coefficient from a linear-log and linear model fit at the species level, while limitation thresholds reflect the *x*-intercept of each regression line. The significance between the slope and intercept of each line was evaluated using an analysis of variance (Supplementary Table 3). Gray areas around lines indicate 95% confidence intervals. *P* values in each figure correspond to the results of a two-sided ANOVA for a comparison between the line of best fit for growth and photosynthesis, respectively. Although the lines were significantly different for *J. monosperma*, this suggests photosynthesis stopped significantly sooner than did growth. A separate analysis of needle, twig, bole, and root NSC for June only can be viewed in Supplementary Figs. 1 and 2.

us from detecting a significant increase in sugar concentrations in these tissues. This may explain the significant decrease in leaf (slope = −0.26, *P* < 0.05) and twig (slope = −0.70, *P* < 0.05) NSC in *P. edulis*. There was nearly a 1:1 conversion of *P. edulis* root starch (slope = −0.6, *P* = 0.08) into sugar (slope = 0.56, *P* = 0.06) leading to no change in root total NSC with decreasing growth (slope = −0.008; *P* = 0.9). These results reinforce our previous observation (see Fig. 3) that NSC does not increase when drought limits the growth of trees in a semi-arid woodland.

During this study, both species displayed a clear coordination of growth, photosynthesis, and carbon storage in response to drought (Figs. 1–3). To explore how these traits interact simultaneously, we used a principal components analysis (PCA) to project our variables ($\psi_{pd}$, canopy starch, canopy sugar, growth, and photosynthesis) onto a two-dimensional subspace. More than 70% of the variance in the data spanning the entire growing season could be explained using just two principal components (PC1 = 47.15%, PC2 = 27.58%; Fig. 5; Supplementary Table 5). PC1 reflects the antagonistic effect of $\psi_{pd}$ on growth and

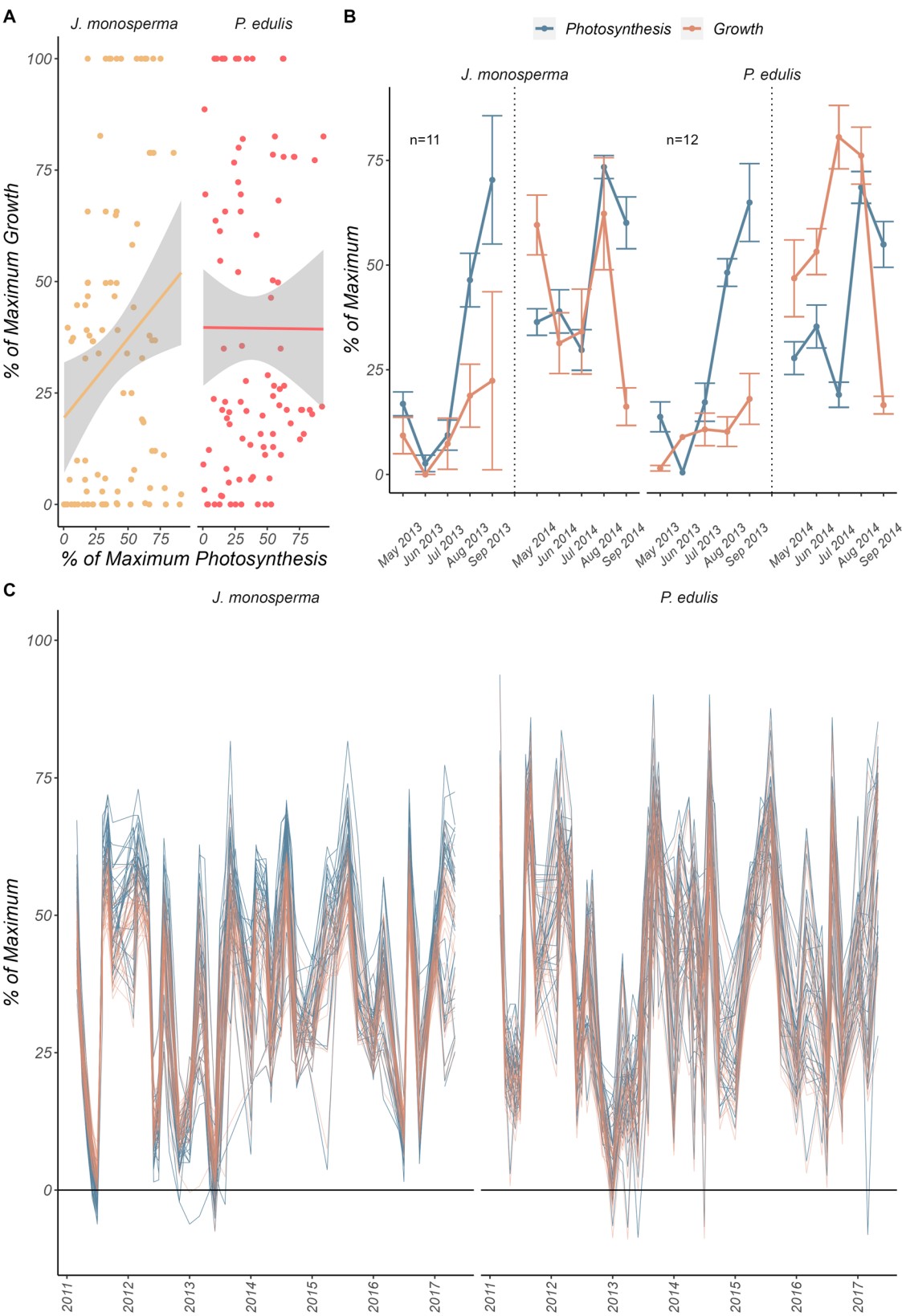

photosynthesis (Fig. 5), and the increase in tree sugar concentrations observed in Fig. 3. Along PC1, co-limitation of growth and photosynthesis was observed. PC2 however, illustrates the independence of growth and photosynthesis with both variables moving in opposite directions. While most of the variance in the data was explained by just two principal components (74.73%; Fig. 5A), this contrast between

growth and photosynthesis was persistent throughout PC2–PC4 (>90%; Fig. 5A). Remarkably similar patterns were observed when June data were analyzed on its own, to include NSC components from the bole, roots, and the canopy (Supplementary Fig. 4).

To further quantify the relationship between growth and photosynthesis in these two species, we calculated the inner product

**Fig. 2 | Synchrony between growth and photosynthesis depends on timing and the environment. A** Growth and photosynthesis are plotted against each other, showing a lack of relationship between the two variables in *P. edulis* (two-sided *F* test, $F = 0.0007$, $P = 0.9$, $R^2 = -0.01$, $n = 94$) but significant correlation in *J. monosperma* (two-sided *F* test, $F = 6.556$, $P = 0.012$, $R^2 = 0.05$, $n = 98$). Though significant for *J. monosperma*, the low $R^2$ value suggests a latent driver of underlying variance (e.g., water limitation). **B** Seasonal variation in growth and photosynthesis suggest the two variables respond in tandem to seasonal variation but diverge toward the end of the growing season. In early fall, photosynthesis remains high, while growth converges to zero consistent with phenological patterns. Vertical dotted lines in (**B**) separate 2013 data (left) from 2014 data (right). **B** is shown for visualization purposes only and thus, no statistical test was run. Nonetheless, $n = 11$ for *J. monosperma* and $n = 12$ for *P. edulis* in both years. **C** A time-series of % of maximum growth (red) and photosynthesis (blue) estimated from measurements of predawn water potential at MDB (25 years) and SUMO (6 years) show that complete cessation of growth and photosynthesis is rare. The dramatic crash in growth and photosynthesis for *P. edulis* at MDB in 2003 indicates a drought-induced mortality event, after which new trees were selected. Although very low $\psi_{pd}$ sufficient to close stomata and stop growth are relatively rare, the 2003 mortality event highlights the importance of rare and extreme events as a driver of tree mortality. **A** Points represent tree-level observations of growth and photosynthesis from May to September 2013 and 2014 at SUMO. Gray areas around lines in (**A**) indicate 95% confidence intervals. **B** Points represent average growth and photosynthesis (across individuals within species) for each month of the growing season at SUMO, while error bars represent standard error. **C** Lines represent individual tree $\Psi_{pd}$ measured monthly through time at MDB (left) and SUMO (right). To estimate the % of maximum growth and photosynthesis, we used the linear-log model from Fig. 1.

of the variable vectors in our data, determined as: Growth · Photosynthesis = |Growth| × |Photosynthesis| × cos($\theta$) and solved for $\theta$. Here, $\theta$ represents the angle between two vectors, where a value approaching 90° indicates two vectors are orthogonal. The angle between growth and photosynthesis was 89°, confirming the independent yet coordinated response of each variable to declining $\psi_{pd}$. Starch fell within the angle between growth and photosynthesis, suggesting carbon storage increases when conditions are favorable to support high levels of growth and photosynthesis (Fig. 5). This is consistent with the strong seasonal patterns we observed in NSC, which reflect phenological patterns of carbon supply and demand[22,23]. Although this may provide some evidence against the view of competing sinks and allocation hierarchies in favor of a more integrated view of plant carbon allocation, we are careful in our interpretation given the absence of molecular data[21].

In this study, we used two co-occuring conifer species growing in a semi-arid ecosystem to address two questions relating to the drought response of trees:: (1) Does growth stop before photosynthesis? and (2) Does NSC accumulate during periods of reduced growth? Central to our understanding of plant carbon allocation is the view that starch accumulates during periods of reduced growth[1,5,24,25]. We demonstrated that growth and photosynthesis decline concomitantly in response to drought in both *J. monosperma* and *P. edulis*. The extrapolated data from $\psi_{pd}$ to growth and photosynthesis suggests that drought conditions severe enough to drive this sink-source co-limitation may be rare in this ecosystem (Fig. 2 and Supplementary Fig. 11). While this suggests that carbon storage may have increased more often than it decreased, direct and simultaneous measurements of $\psi_{pd}$, growth, photosynthesis, and NSC are needed. Moreover, this study only evaluated radial growth changes. While we recognize that elsewhere on these trees growth could have persisted, radial growth is often the first to stop in response to drought. Even if growth had continued elsewhere on the plant, the rapid decline in photosynthesis still prevented carbon storage.

Large-scale field experiments and long-term observational datasets such as these are often used to parameterize models that predict ecosystem responses to climate change[16,26–28]. It is these rare but extreme events that often matter most in determining whether plants live or die[11], and are thus often the subject of such models. Most models[16] still reflect the theory that growth stops before photosynthesis[1,3], that carbon storage increases following cessation of growth[1,5], and that photosynthesis drives growth[29]. Our results suggest that NSC does not increase during drought in all species. It remains unclear whether the dynamics we observed in this study apply to trees in other ecosystems, with distinct climates, evolutionary histories, or functional types. To our knowledge, this study, however, provides the first evidence that growth does not always stop before photosynthesis, that NSC does not accumulate in all tree species during drought, and that growth does not depend on photosynthesis. Future work should aim to test these observations in other species in a variety of environments.

We offer an alternative perspective on how plants use carbon (at least in terms of growth, photosynthesis, and carbon storage) that views growth and photosynthesis as largely independent processes influenced by complexes of molecular, biophysical, and chemical regulators. Beyond growth and storage, other sinks, such as defense and respiration, play significant roles in plant carbon balance and have implications for their sensitivity to biotic and abiotic stress[9,11,25]. It is critical that future research move beyond measurements of growth and NSC as simple proxies for plant carbon balance during drought. Until vegetation models account for the complexities of plant carbon dynamics, sink-source co-limitation, and sink multiplicity, we will remain challenged in predicting the response of earth's largest terrestrial carbon sink to climate change[30].

## Methods

### Site description and experimental design
Due to the fusion of data from multiple sites, this methods section is structured such that each sub-header begins with a discussion of SUMO and then MDB. Where a measurement did not occur for trees at either the SUMO or MDB sites, it is explicitly mentioned. Since both sites are near each other and occupy similar habitats, the basics of the site description apply to both sites.

The Los Alamos Survival/Mortality (SUMO) experiment[19,31–33] and MDB site[17,34] have been described in detail previously by others. Briefly, the SUMO and MDB sites are located near Los Alamos, New Mexico USA, at elevations of 2150 m and 2140 m, respectively, in piñon–juniper woodland just below the *Pinus ponderosa* forest ecotone. *Pinus edulis* and *Juniperus monosperma* dominate both sites, although scattered individuals of *Quercus gambelli*, *P. ponderosa*, *J. deppeana*, and *J. scopulorum* can also be found at the SUMO site. A volcanic tuff parent material sits below the Hackroy clay loam soils that are found at both sites and can be found at depths ranging from 40 to 80 cm. The growing season occurs between April and October. Average 30-year temperature and precipitation were 10.1 °C and 360 mm, respectively. At both sites, trees growing naturally in the field (i.e., not planted but naturally recruited) were selected for observation and experimental study.

At SUMO, below-canopy precipitation removal structures and open-top heating chambers were installed during June 2012. A total of 64 individuals of *P. edulis* and *J. monosperma* (32 trees per species) growing in the ground were selected and placed into one of five treatments (5–7 trees per species in each), however due to a lack of growth data, only four treatments are considered in this paper. The ambient treatment consisted of trees exposed to ambient temperature and precipitation. The heat treatment was implemented by placing open-top chambers around selected trees to create an average increase of 4.8 °C above ambient temperatures. Drought trees were exposed to ambient temperatures within a precipitation removal structure that diverted ~45% of precipitation away from these trees.

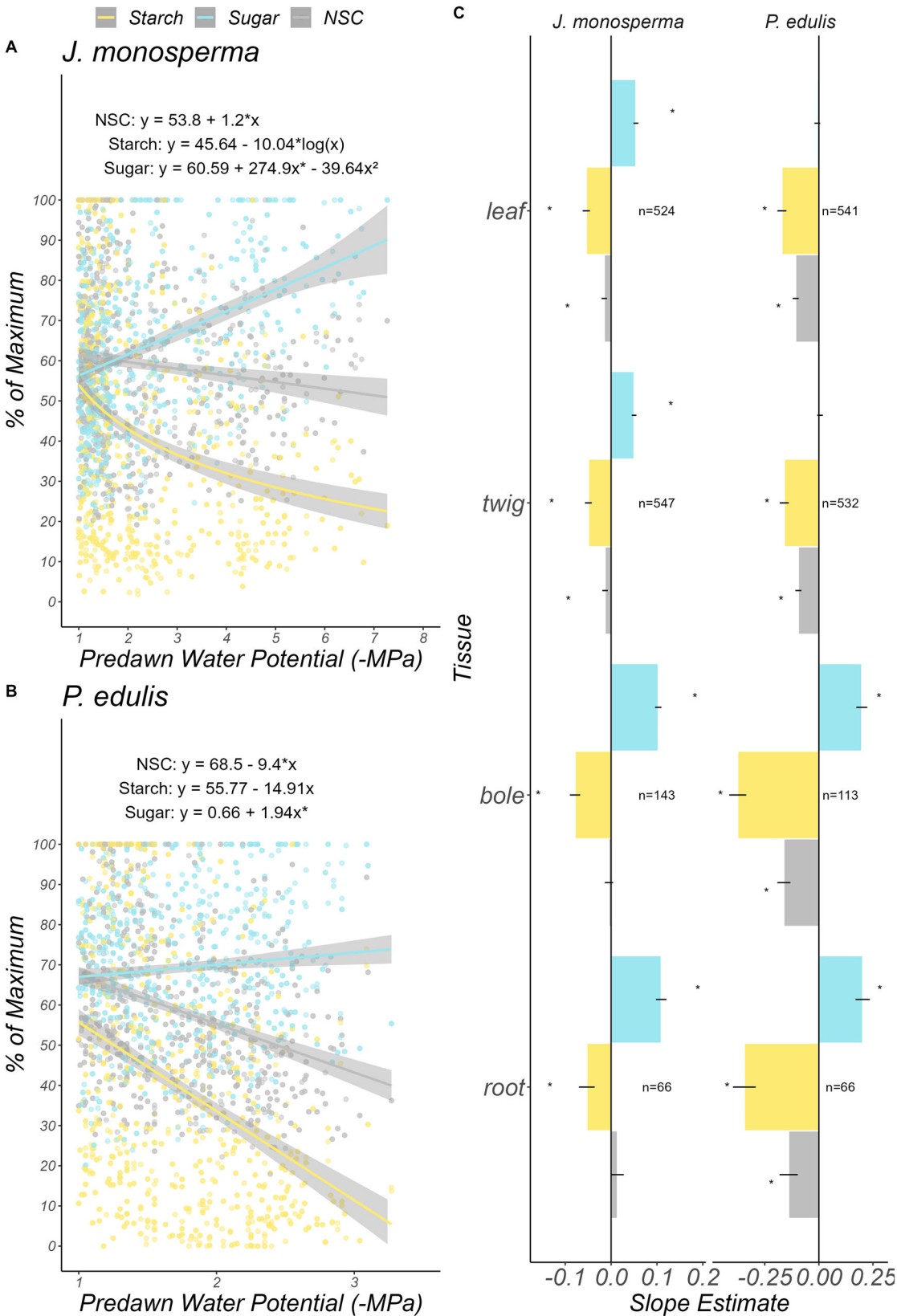

Heat + Drought trees were exposed to both the 4.8 °C temperature increase and the precipitation removal. Continuous measurement of site climatic conditions using two weather stations, in addition to within-chamber measurements, allowed control of chamber conditions using heating and air-conditioning units. In general, all measurements were made on the same trees such that comparisons of growth, photosynthesis, water potential, or NSC are robust. However, every parameter was not measured on every tree. Where all measurements in an analysis are not present for every tree in the study, those trees are excluded from that analysis. All trees assess in this study at SUMO had measurements of growth, photosynthesis, water potential, and NSC.

**Fig. 3 | NSC does not increase in *J. monosperma* and *P. edulis* during drought.** As $\psi_{pd}$ becomes more negative, both species reallocate carbohydrates from the canopy to the roots (**C**). In *J. monosperma*, this is likely to increase water supply to the roots and allow stomata to remain open. Additionally, *J. monosperma* increased the sugar concentrations of apical tissues, consistent with osmotic adjustment under drought. In *P. edulis*, osmotic adjustment of apical tissues did not occur as stomata closed relatively early. Instead, increased sugars in the bole and roots likely supported metabolism during extended periods of stomatal closure. These tissue-specific patterns are reflected by the unweighted averages shown in (**A, B**). Points in (**A, B**) are observations of canopy sugar (blue points and line) and starch (yellow points and line). Lines in (**A, B**) represent point estimates of regression coefficients for the line of best fit, determined using the Akaike Information Criterion (Supplementary Table 11). **C** Bars represent point estimates of the regression coefficient for each NSC component and tissue against predawn water potential. Gray areas around lines in (**A, B**) and error bars in (**C**) indicate 95% confidence intervals. A separate analysis of needle, twig, bole, and root NSC for June only can be viewed in Supplementary Fig. 3. Tissue-specific regressions can be found in Supplementary Figs. 8–10.

At the MDB site, five trees each of *P. edulis* and *J. monosperma* were selected for long-term monitoring in March 1992, and two additional *P. edulis* trees were added in 1994. In 2003, all seven measured *P. edulis* trees died from drought and bark beetle attack, and five surviving replacements were selected in 2004. Measurements from one tree were switched to another in 2008. Several *J. monosperma* were added to measurements in subsequent years: five in 2007, and three in 2015.

### Radial growth measurements and calculations

At SUMO, tree radial growth was measured as outlined by Manrique-Alba et al.[33]. Briefly, from May through September in 2013 and 2014, a linear variable displacement transducer (LVDT) was attached to the upper bole of 11 individuals of *J. monosperma* and 12 individuals of *P. edulis* using a rectangular frame that was attached directly to the tree using screws. Trees were selected from Heat, Heat + Drought, Drought, and Ambient treatments. Dead bark was gently removed from the site where the sensor contacted the tree, though a thin layer was left as to protect the phloem and prevent water loss. Data from the LVDT is a relative metric of change in diameter over time. Upon installation, each LVDT was set to "zero" the night of the first day of the measurement period before any growth occurred. This established the reference point from which measured growth would deviate. Because diurnal fluxes of stem diameter make calculating growth difficult, we considered growth initiation to have occurred when the maximum stem diameter exceeded that of the previous day's maximum, for each tree. When maximum stem diameter did not exceed the previous day's maximum, we considered growth to have stopped. No direct measurements of growth were made on trees at the MDB site.

### Water potential and photosynthesis

At SUMO, xylem water potential and foliar gas exchange were determined for 11 individuals of *J. monosperma* and 12 individuals of *P. edulis* (2011–2017). Two twig samples were collected every three months from each tree before sunrise and the xylem water tension was measured using a Scholander pressure chamber (PMS Instruments, Albany, OR, USA). The level of water stress for each tree was quantified as the average predawn water potential ($\psi_{pd}$) of both stems. At MDB, xylem water potential was measured every month between 1992 and 2016. Measurements were made on 5–6 individuals of *J. monosperma* from 1992 to 2012, and 11–14 individuals of *J. monosperma* from 2013 to 2016. Between 1992 and 2016, water potential measurements were made on 5–7 individuals of *P. edulis*.

At SUMO only, net photosynthesis and gas exchange were measured on the south-facing, sun-exposed side of each tree using a Li-Cor LI-6400 (Lincoln, NE, USA). Needles from each tree were measured under chamber conditions set of 380ppm CO2, 1500 mol $m^{-2}s^{-1}$ light-saturating photosynthetic photon flux density (PPFD), temperatures between 20 and 25 °C, and 0% relative humidity. These conditions closely matched those of the outside environment, where temperatures ranged from 13 to 30 °C and 750 to 1800 mmol $m^{-2}s^{-1}$ PPFD. After two minutes of steady-state gas exchange, measurements were recorded, and needle samples were collected to determine leaf area. Gas exchange data was corrected using leaf area measurements made on a Li-Cor LI3100C area meter. No gas exchange measurements were made on the trees at MDB.

### Non-structural carbohydrates

Beginning on March 14, 2012, and ending on October 13, 2016 at SUMO, leaf and stem (twig; phloem and bark) tissue samples were collected for each tree four times each year to capture seasonal changes associated with spring dormancy break, mid-summer drought, monsoon wet-season, and post-monsoon dry-down. Stem samples were from recent growth, dated from 0 to 5 years old for *P. edulis*. Bole and root samples (also including bark) were collected using an increment borer once per year during the dry season, in June. All samples were collected between 11:30 and 13:00, mitigating any influence of diurnal variation in NSC on our measurements. Upon collection, samples were placed into liquid nitrogen and transported to the lab in dry ice. Samples were kept stored at −70 °C until analysis, when they were microwaved for 5 min at 800 W and placed in a drying oven for 48 h at 65 °C. All samples were ground using a ball mill and woody tissues were preground using a Wiley Mini-Mill. To assay NSC, we used the protocol outlined by Dickman et al.[35], as developed from the methods of Hoch et al.[36]. This method has been verified to produce reasonably accurate and precise measurements of NSC, defined as glucose, fructose, sucrose, and starch[37]. Approximately, 12 mg of finely ground sample was placed into a deep-well plate with 1.6 mL deionized water and placed into a 100 °C water bath for 1 h. An NAD-linked enzymatic assay was used in combination with spectral assessment at 340 nm for NSC quantification. To analyze NSCs at the whole-tree or canopy scales we averaged NSC concentrations from each respective tissue. For example, to calculate canopy NSC, sugar, and starch, we averaged the NSC from stem and needle tissues. A similar approach was used to estimate whole-tree NSC for June. No NSCs were measured on the trees at MDB.

### Data analysis

At SUMO, prior to data analysis, outliers in the growth and photosynthesis data that exceeded 3x the mean cook's distance were removed[38]. To begin our analysis, we evaluated the response of growth and $A_{net}$ to declining $\psi_{pd}$. Because of the disjointed sampling frequency of $\psi_{pd}$ and stem radial growth, we normalized the $\psi_{pd}$ at which growth and $A_{net}$ stopped for each tree. We first took the sum of all daily growth between $\psi_{pd}$ measurement intervals as the total monthly growth for each tree. To account for tree-specific variation in growth rates, maximum growth was first determined at the individual-tree level and within-tree growth measurements being expressed as a percentage of that tree-level maximum. This approach is consistent with previous work on this subject, where growth and photosynthesis were expressed on relative scales[3]. At the species level, a linear-log regression was run (due to limited number of observations at the individual level), identifying the response of growth to declining $\psi_{pd}$ (Supplementary Fig. 1A, C). Limitation to photosynthesis was determined similarly, first with percent of maximum photosynthetic rates established at the tree-level and a linear-log regression to evaluate species-level response to drought (Supplementary Fig. 1B, D). The regression model used to describe the relationship between $\psi_{pd}$ and

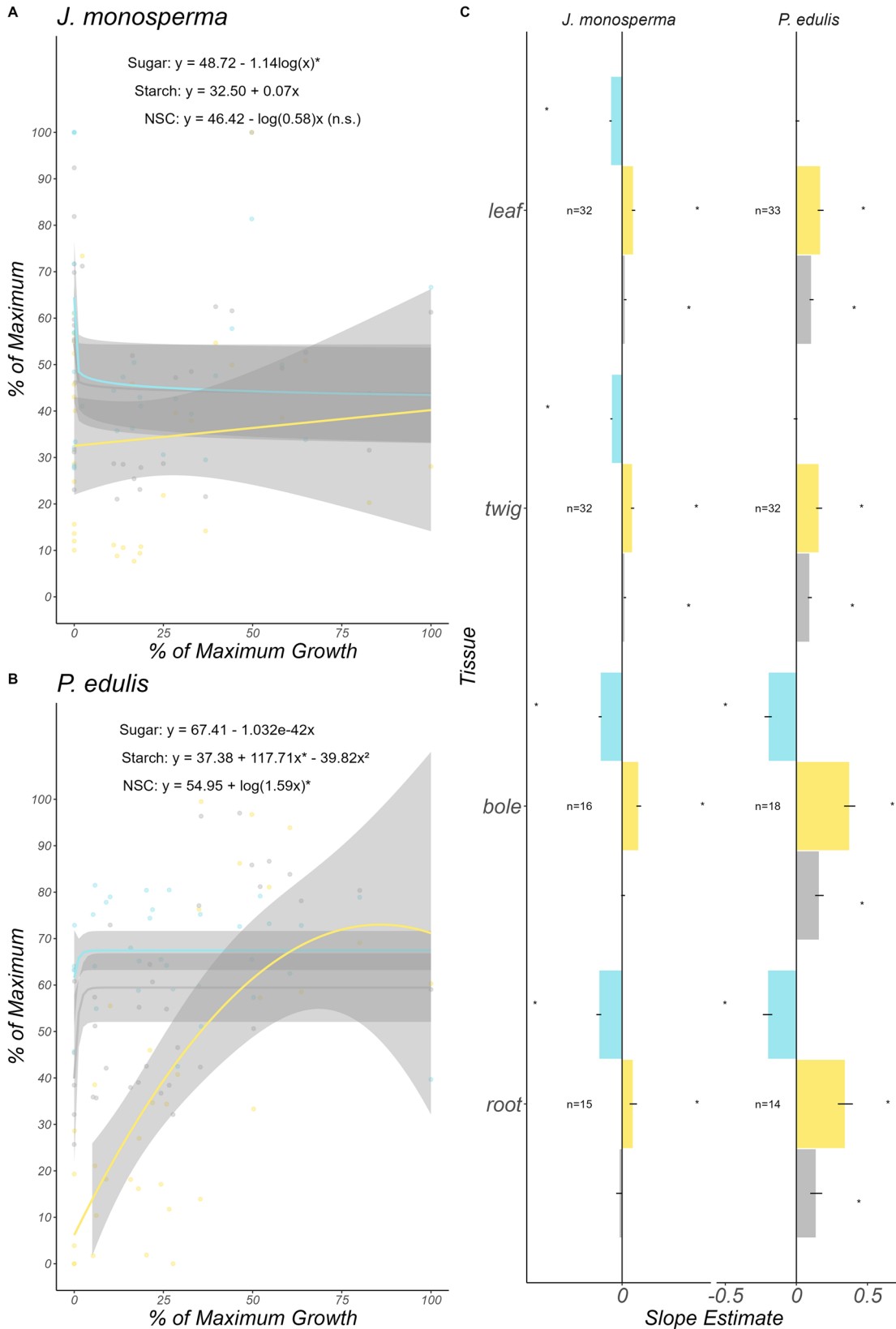

**Fig. 4 | NSC does not increase in *J. monosperma* and *P. edulis* as growth declines.** As growth becomes limited, NSC, starch, and sugar concentrations do not significantly change in *J. monosperma* (**C**). In *P. edulis*, a significant decline in NSC, driven by a decline in starch, may drive the significant increase in sugar concentrations in the bole. These tissue-specific patterns are reflected by the unweighted averages shown in (**A**, **B**). Points in (**A**, **B**) are observations of canopy sugar (blue points and line) and starch (yellow points and line). Lines in (**A**, **B**) represent point estimates of regression coefficients for the line of best fit, determined using the Akaike Information Criterion (Supplementary Table 11). **C** Bars represent point estimates of the regression coefficient for each NSC component and tissue against growth. Gray areas around lines in (**A**, **B**) and error bars in (**C**) indicate 95% confidence intervals. Tissue-specific regressions can be found in Supplementary Figs. 12 and 13.

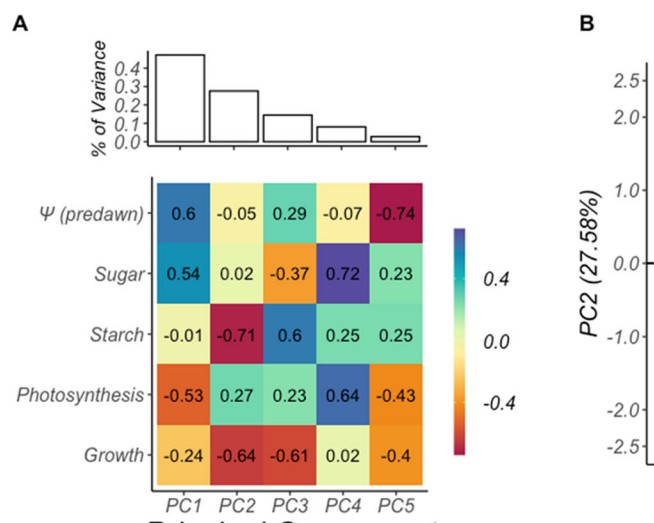

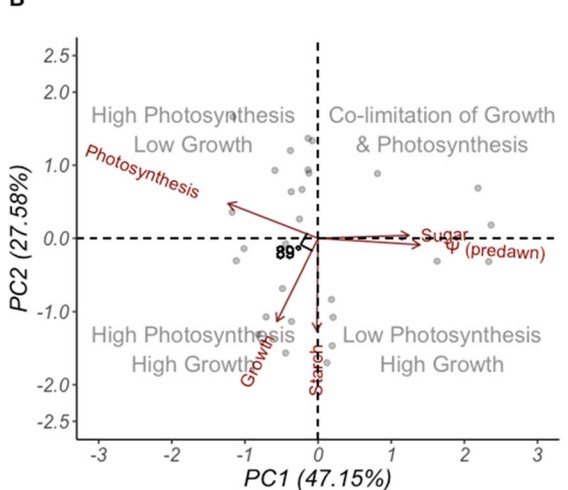

**Fig. 5 | $\psi_{pd}$ couples photosynthesis to growth.** During drought, carbon availability is not the limiting process influencing growth rate. Drought-induced reductions in cell turgor independently limits growth and photosynthesis, explaining both sink-source co-limitation and the lack of correlation between growth and photosynthetic rates. Results of the principal components analysis on carbon supply and demand variables reveals that water availability ($\psi_{pd}$) limits growth and photosynthesis and not carbon availability. PC1 explained 47.15% of the total variance in the data, and together PC1 and PC2 explain 74.73% of the total variance in the data. Along PC1, $\psi_{pd}$ becomes more negative, drive sink-source co-limitation. Sugar concentrations increase as starch is depleted to support the metabolic and osmotic needs of both species. Growth and starch concentrations vary the most along PC2, reflecting the temporal patterns of plant phenology and starch accumulation (Supplementary Fig. 6). **A** A scree plot showing the amount of variance each principal component accounts for, and the eigenvectors associated

with each principal component are shown. **B** Biplot of PC1 and PC2 are shown. Photosynthesis and growth contrast along all principal components except PC1 (water stress axis) and PC5 (which explained <10% of the total variance). This result stands in contrast to what theory and models would predict[1,5,16,24]. Based on theory, carbon demand drives carbon supply[29] and starch accumulates inversely to growth[1,5,24]. Comparison between the inner product of $\overline{Photosynthesis}$ and $\overline{Growth}$ is indicated by $\theta = 89°$; suggesting the two variables are orthogonal to each other. All variables (expect water potential, which is shown in absolute terms) are calculated as a percentage of the maximum observed for each tree. Each point indicates a specific observation where all variables overlapped (i.e., were observed for the same sampling month). Starch and sugar shown here are canopy averages only, due to the low sampling frequency of bole and root NSC measurements. A separate analysis across the entire growing season (NSC includes only needle and twig) can be viewed in Supplementary Fig. 5.

growth or photosynthesis, respectively, was determined using AIC. The best performing growth model was a linear-log model, while photosynthesis performed best when using a polynomial model (see Supplementary Tables S8 and S9). This was true for both species. Since our aim was to evaluate carbohydrate dynamics following complete cessation of growth and photosynthesis, as well as along the trajectory to complete cessation, the polynomial model was rejected in favor of the second-best-fitting model, the linear-log model ($\Delta AIC < 7$). Although trees were placed into different treatments for the duration of the SUMO study, we used cross-treatment fits to expand the observed range of $\psi_{pd}$, making our analysis more robust. Thus, we chose to evaluate differences in thresholds and NSC dynamics on all trees at the species level, increasing our sample size from ~3 trees per species per treatment to 11 individuals of *P. edulis* and 12 individuals of *J. monosperma*.

Statistical analyses were conducted using base R[39]. We began by evaluating the goodness-of-fit of several linear and non-linear regression models on the relationship between growth and photosynthesis with $\psi_{pd}$ (Supplementary Tables S8 and S9). After fitting several models to the data, goodness-of-fit was determined using Akaike's Information Criterion and the model with the lowest dAIC was selected. Because we were interested in models where the regression line crossed the x-axis (a necessary condition for finding points of cessation), model selection was further filtered as the model with the smallest dAIC <7[40]. The result was that polynomial fits were ultimately excluded even if they fit the data best.

To test for differences in the points of cessation and slopes of growth and photosynthesis with $\psi_{pd}$, we used an analysis of variance on the interaction of each variable (Supplementary Table 3) in the aforementioned model and analysis of variance with growth and

photosynthesis as response variables. Although growth and photosynthesis vary continuously with $\psi_{pd}$, this approach allowed us to identify a threshold beyond which plant carbon dynamics could be evaluated. To test the hypothesis that growth limitation induces an increase in NSC, we began by determining relative values of NSC, sugar, and starch for each tree. As with photosynthesis and growth data, we determined the tree-level maximum NSC, sugar, and starch, and expressed all subsequent values as relative to that. Regressions were run at the species level by pooling individual-tree relative values into a species-level dataset. We evaluated a set of candidate models with NSC, sugar and starch as a response variable, and either % of maximum growth or $\psi_{pd}$ as a predictor variable. Like our analysis of growth and photosynthesis, sugar and starch concentrations were likewise expressed as a percentage of the maximum for each tree. This approach enhanced our ability to account for differences between trees, detect relative changes in NSC concentrations, and enhanced our qualitative analysis. The candidate models we tested were polynomial, negative exponential, linear, and linear log. We evaluated the best-fitting model for each species, component, and predictor variable separately (see Supplementary Tables S6 and S7 for AIC results; Fig. 3 shows best-fitting models and their mathematical form).

The amount of time that trees spent at different rates of growth and photosynthesis relative to their maximums was of particular interest, with respect to generalizing the effects observed here to how trees might respond under future warming. For both species, at SUMO, we calculated the average frequency (expressed as a % of the study length) of growth and photosynthesis limitation.

At MDB, growth and photosynthesis were not directly measured. Therefore, we used the linear-log models fit to the $\psi_{pd}$ versus growth and photosynthesis data at SUMO, to inverse-predict the estimated %

of maximum growth and photosynthesis at a given $\psi_{pd}$, in 10% intervals (Supplementary Fig. 7). Then, at both sites, we calculated how often each species spent at each interval and used a Kolmgorov–Smirnov test to evaluate differences in the distribution. If the trajectory toward cessation differed between growth and photosynthesis, we would expect the distributions to be significantly different.

Finally, to better understand the interdependence of growth, photosynthesis, sugar, and starch to drought, we used a principal components analysis only on the data from SUMO. The apparent coupling of growth and photosynthesis to declining $\psi_{pd}$ (Fig. 1) was of particular interest given the widely cited conjecture that photosynthesis drives growth[29,41]. This analysis leveraged photosynthesis, growth, NSC, and $\psi_{pd}$ data, yet was limited by the infrequency of the growth measurements. Still, the dimensionality of the dataset was not outweighed by the limitations of the sample size, and we were able to analyze the centered and scaled data using a correlation matrix of the data (see Supplementary Tables S4 and S5 for PCA results). Orthogonality between variables was measured using the dot-product of each vector, expressed as $\theta$, effectively measuring the independence between two vectors[42]. Because the estimates of growth and photosynthesis at MDB were made from $\psi_{pd}$ data, we could not evaluate whether growth and photosynthesis were independent in these trees. Therefore, we did not conduct a principal components analysis using any data from MDB. All data analysis was conducted using R (version 4.2.1) and the functions lm(), princomp(), and anova().

### Sensitivity analysis
In addition to the above analysis, we replicated the main results using alternative maximum values. The result of this further analysis is included in the accompanying supplementary file titled Sensitivity Analysis. The aim of the sensitivity analysis was to test the robustness of our results to the choice in maximum value, against which all subsequent measurements for each individual tree were set relative to. This approach facilitated species-level analyses on individual-level measurements. Our sensitivity analysis confirms the above-mentioned results and suggest that the choice of maximum value had no effect on driving the patterns we describe above.

### Reporting summary
Further information on research design is available in the Nature Portfolio Reporting Summary linked to this article.

## Data availability
SUMO data is available at https://doi.org/10.15485/1440544.

## Code availability
All R code can be found on Github at https://zenodo.org/badge/latestdoi/310685648.

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

## Acknowledgements

The Los Alamos Survival-Mortality Experiment (SUMO) was funded by the US Department of Energy, Office of Science, Biological and Environmental Research. R.A.T., A.M.T., and H.D.A. were supported by the NSF Division of Integrative Organismal Systems, Integrative Ecological Physiology Program (IOS-1755345, IOS-1755346). R.A.T. was also supported by the NSF Graduate Research Fellowship Program. H.D.A. was also supported by the USDA National Institute of Food and Agriculture (NIFA), McIntire Stennis Project 1019284 and Agriculture and Food Research Initiative award 2021-67013-33716. C.G. was supported by the Swiss National Science Foundation (310030_204697).

## Author contributions

All authors contributed to interpreting the analyses and editing the paper. R.A.T. wrote the manuscript, analyzed the data, and designed the figures. H.D.A. and R.A.T. developed the hypotheses. H.D.A., D.D.B., A.D.C., L.T.D., C.G., A.M.A., A.M.T., and N.G.M. collected the data. N.G.M. designed and led the SUMO project. D.D.B. designed and led the long-term MDB observations. H.A.D., D.D.B., A.D.C., L.T.D., C.G., A.M.A., A.M.T., N.G.M., M.G.R., and D.M.P. contributed significantly to data analysis and writing and preparing the manuscript.

## Competing interests

The authors declare no competing interests.
