## [Peer Review File · Nature Communications]

REVIEWER COMMENTS

Reviewer #1 (Remarks to the Author):

Thompson et al. put together a very impressive conglomeration of long-term datasets measuring water potential, radial growth, photosynthesis and carbohydrate storage. Data are collected in the American Southwest and are from a combination of in situ measurements and experimental drought treatments. Using their data, they explore the long-debated source versus sink limitation in storage formation, with respect to drought. They ask the questions:

1. Does growth stop before photosynthesis in response to declining water potentials?
2. Does carbon storage increase as growth declines with more negative water potentials?

Both of these suppose sink limitations, in which growth is more sensitive to environmental stress than photosynthesis, thus carbon continues to be assimilated and added to storage, rather than being put toward growth when under stress. Interestingly, the study found that (1) photosynthesis ceased before growth did in both pine and juniper, suggesting the foundation of the sink limitation hypothesis may not actually be sound. Next, they found that (2) starch stores decreased steadily in response to drought, rather than increase as they predicted.

While I find the first finding very interesting and backed by an impressive dataset, the majority of the paper is focused on the second finding and the question of storage, which I have several major issues with. Overall, the dataset is impressive, but the framing of the data and the analysis of the storage data in particular leaves a lot to be desired.

Major Points:

1. The recurrent refrain and framing of the paper is that starch (which they claim is the storage molecule) increases following drought and this idea is well known and pervasive in the field (L35-36, L56-57, L177-178). The authors then set out to disprove this with their dataset. I have a two-part issue with this idea:

- a. Starch as the only storage molecule of interest I think is inappropriate here. While starch is thought of as long-term storage because it is insoluble, both sugars and starches (combined as NSC) are being put away by plants for future use. Whether it is stored as sugar or starch varies on daily to annual timescales – sugar is regularly converted back and forth into and out of starch daily (Gersony et al., 2020; Tixier, Orozco, Roxas, Earles, & Zwieniecki, 2018), seasonally (Martínez-Vilalta et al., 2016), and in response to environment (Thalmann & Santelia, 2017).

- b. This flexibility of plants to convert stored NSC back and forth is very well established, particularly in response to drought. Many studies have demonstrated that following drought, starch is broken down into sugar for a number of regulatory functions (eg. Thalmann & Santelia, 2017). Thus, I'm not sure how you come up with Fig 1A, I think your findings illustrated in Fig 1B & C are what I would have predicted for starch given the literature. Indeed, none of the papers you cite to support your

assertion that starch accumulates following drought actually say this. Körner et al. 2003 only presents total NSC data and while he shows that total NSC increases following drought, if you return to the original dataset, you find that starch is broken down. Similarly, Wiley and Helliker 2012 only refer to total NSC, starch is in fact never mentioned in the paper, and Monson et al. 2021 is a review of growth-defense tradeoffs, where storage is really just peripheral to the paper.

2. Relatedly, only examining starch and sugar separately and as percentages of their maximum makes interpretation very difficult. While it makes comparisons to other measurements easier, it also may be hiding some significant patterns. For one, it makes starch and sugar look like they are equitable in the amount of energy they may provide. However, whether starch is 2% or 80% of total NSC would drastically alter how one views your findings. Storage should really be represented here by total NSC, or that metric should at least be included in your analysis to truly understand what is happening with storage post drought.

3. Statistics could be much better reported throughout the text, figures, and tables. I have more minor comments below that indicate where it is a particular issue, but an eye throughout to incorporating the results of statistical tests to back up your findings in text and in figures is needed. In general, it would be good to report slope estimate, sample size, and p-value in text to back up your statements. Tables should also be more standardized in their reporting - in some cases like table S2 all relevant information is included, but then in table S6 parameter estimates are not provided.

Minor Points:

1. The writing would benefit if the terms source/sink limitation were defined upfront (before L86) as used more consistently throughout, as that is what they are ultimately testing.
2. L142 – provide slope, give N and p-val as well.
3. L94 – growth did not stop before photosynthesis in both species, or just juniper? Confused because you later say “in contrast”, suggesting species are doing different things.
4. L100 – I think a little more detail on the two species and why they are expected to respond to drought differently would go a long way to understanding the results.
5. L 224 – are they planted in pots or in the ground?
6. L 268 – At what date did collection of tissue for NSC end?
7. Fig 1. You show the result of three separate regressions as equations, but report 1 p-value. What does this correspond to?
8. Fig. 2A – While they appear uncorrelated, you should back this up with statistics.
9. Fig. 2B – would be helpful to highlight the growing season on graphs for these species and show that there is break in measurements between years.
10. L108-110; Fig 2C – I am a bit confused, you make the argument in 2A that photosynthesis and growth are uncorrelated, but then they look remarkably coordinated in 2C. I’m not sure entirely how

you get from 2C complete dataset to 2A. How are you summarizing data here? It says tree-level observations, but then are you averaging over all months too?

11. I don't have issue that you are drawing on multiple datasets from different individuals, but I think it could be more clear and transparent in the text what comes from where and when. Possibly a table or timeline graphic could be helpful.

Cited Papers

Gersony, J. T., Hochberg, U., Rockwell, F. E., Park, M., Gauthier, P. P. G., & Holbrook, N. M. (2020). Leaf Carbon Export and Nonstructural Carbohydrates in Relation to Diurnal Water Dynamics in Mature Oak Trees. *Plant Physiol*, 183(4), 1612-1621. doi:10.1104/pp.20.00426

Martínez-Vilalta, J., Sala, A., Asensio, D., Galiano, L., Hoch, G., Palacio, S., . . . Lloret, F. (2016). Dynamics of non-structural carbohydrates in terrestrial plants: a global synthesis. *Ecological Monographs*, 86(4), 495-516. doi:10.1002/ecm.1231

Thalmann, M., & Santelia, D. (2017). Starch as a determinant of plant fitness under abiotic stress. *New Phytol*, 214(3), 943-951. doi:10.1111/nph.14491

Tixier, A., Orozco, J., Roxas, A. A., Earles, J. M., & Zwieniecki, M. A. (2018). Diurnal Variation in Nonstructural Carbohydrate Storage in Trees: Remobilization and Vertical Mixing. *Plant Physiol*, 178(4), 1602-1613. doi:10.1104/pp.18.00923

Reviewer #2 (Remarks to the Author):

In the manuscript entitled "No carbon storage for growth-limited trees" Thompson et al. used a long term monitoring of predawn water potentials dataset from a juniper-pine woodland (MDB) and a 7 years monitoring of predawn water potentials, radial growth, leaf photosynthesis and NSC concentrations in a rainfall exclusion experiment (SUMO) to evaluate carbon source-sink dynamics under water availability fluctuations. The authors tested the hypotheses that: growth stops before photosynthesis in response to declining predawn water potential (i.e. proxy of water availability in the soil for plants); carbon storage increases as growth declines due to decrease in predawn water potential. The authors conclude that, for the two conifer species analyzed, co-limitation of growth and photosynthesis is rare, except under extreme drought. Different patterns in growth and photosynthesis decline with decreasing predawn water potentials were observed in the two conifer species, in line with their different drought stress tolerance behaviours. For the same reasons, also starch and sugar patterns with drought progression were different.

The manuscript is well written. I agree with the authors' conclusion that co-limitation of growth and photosynthesis would be the result of a coordinated but independent response to drought. However, under non-drought conditions, it is known that along the growing season several factors determine independently the two processes. E.g. tree phenology operates separately on growth and photosynthesis (e.g., in the end of summer growth decreases or cessates, while photosynthesis still could be high compatible with the evergreen habit and to high availability of water due to abundant precipitation). This justifies what observed in fig. 2B in the end of 2013 and 2014 growing seasons. For example, the fact that growth and photosynthesis are uncoupled in the second part/end of the summer (when water availability is high/no drought occurs) allows to increase the storage of NSCs. In addition, photosynthesis and generally leaf gas exchanges depend not only on predawn water potentials but also on VPD. For all these reasons, the lack of relationships between growth and photosynthesis shown in Fig. 2A, given the dataset used (which pools data from the whole growing season), is not surprising.

Looking at methods, sampling and data handling for NSC should be made more clear:

1- The authors present an NSC dataset, showing starch and sugars dynamics. However, given that they took samples from different tree organs in each tree individual, it is not clear how the authors used the different NSC samples collected. In particular, it is not clear how the authors could pool leaf and stem NSC in a same dataset to explain NCS variation dynamics. How did they scale them up to the whole tree level? Leaves and stems can have huge differences in NSC concentrations (and in the total NSC contribution for the plant) and also differ widely in terms of NSC dynamics over time (Hoch et al., 2003). This because leaves are photosynthesis sites and are more affected by photosynthesis dynamics. Stem NSCs are dependent on phloem transport, long term storage, growth etc.. Therefore, if not already done, it could be important to perform the analyses separately in leaves and stems (and also roots, and bole) to check for organ-dependent dynamics. Figg. 3a and 3B show a high variability in starch variation (% of maximum) over water potentials: might this be explained by distinct behaviors among organs?

2-NSC (especially starch) concentrations usually change a lot during the day (especially in leaves, where starch accumulates during the day and is depleted during the night; but also in stems variations can be large, e.g Tixier et al. 2018, Plant phys.). The authors did not specify if samples were taken in a specific time of the day or not. If samples were taken at any time, this could be a major issue in the use of these data, if they do not demonstrate that daily variation in their trees is negligible.

3-The authors should specify what they mean for stem: was it a branch? Of which age (this can highly change NSC concentration)? Did they keep the bulk stem NSC (bark + wood)? This should be specified.

At last, the authors studied two conifer species. Different patterns could be found in e.g. broadleaf deciduous trees, which differ from conifers in many aspects, like e.g. the amount of stored NSCs in the stem, phenology, leaf habit..

Minor comments:

L.98: fig 1 B, not A, right?

L.100: fig. 1C, not B, right?

LI. 120-122: the authors mention values of Ppredawn reached in 2013 and 2014, referring to Fig. 2C (displaying growth and photosynthesis), but there are no values of Ppre-dawn in the figure mentioned. It is also not clear to which experiment the authors refer to with those water potential data: SUMO, MDB, both? Please specify.

L. 183-184: I agree. Belowground compartments under drought, for example, are understudied in this sense..

L. 262: is 0 RH correct?

Reviewer #3 (Remarks to the Author):

The manuscript by Thompson et al. uses two datasets that include measurements of water potential, growth and photosynthesis to assess whether growth and photosynthesis are decoupled under decreasing plant water potential and whether growth decline led to increased NSC storage. A direct test of the decoupling of growth and photosynthesis is rare in most ecosystems and remains an important question for informing vegetation models, especially as climates continue to change. I found the manuscript well written and valued the assessment of the responses to decreasing water potential. I would suggest readdressing your two questions in the final paragraph and providing a clear answer/synthesis of the conclusion to those two questions. I would qualify that your results may be specific to semi-arid gymnosperms and not necessarily applicable to other systems or functional groups.

Despite my general positive view of this manuscript, I found the methods missing simple information and the presentation of the frequency, timing and duration of measurements difficult to follow for each separate dataset. Due to that lack of clarity, it is difficult to assess the validity of the analyses, especially as the response variables are on a relative scale and the calculation of that relative variable can alter the results. Therefore, I strongly suggest the methods are revised with greater detail and clarity (details below).

Detailed points

L95: I believe based on Fig. 1 this value should be 33.3 not 0.33.

L120-122: Typically, averages are given with some metric of variance. Please provide.

L125: What difference? Between the species or years?

L135: Why are figures being presented out of order? I would switch A,B with C,D in figure 3 to maintain order.

L150: Two Bs, delete one.

Methods

It is unclear to me if growth was measured on the same trees that had measurements of water potential and photosynthesis. This is critical for interpreting the robustness of the data and analyses.

In general, I found the methods difficult to follow due to the fact that there are datasets. Would it be useful to reorder the methods into sites and broadly describe what was measured, for what periods and frequency. Then a detailed section on the methodologies of water potential, photosynthesis, etc.

For example,

L251: It is not clear how often measurements were made for SUMO.

L215-216: Monitored for what? How often were trees measured for growth in MDB

L286-287: It is not clear how disjointed the sampling was at MDB. Simple measurement details have been glossed over that are essential to understanding the analyses, especially as all the results are presented on relative to maximum scales.

L289-296: So, a single measurement was used as the maximum for an individual. Would not an average of the top 95% or something similar be more conservative or even the average maximum of all individuals? Please provide more support for the selection of a single value as the maximum or test the effects of alternatives calculations on the results.

L306: How were % maximum NSC determined?

L306-307: Pooling across all treatments? Be more explicit.

L339-341: PCA depends on normally distributed data. Did the data meet this requirement? Were the relative values also used in the PCA?

Dear Reviewers,

We thank you for your thoughtful and helpful comments. We have adjusted the manuscript as requested and respond to each specific comment below. All of our responses are *italicized* and a fully formatted document of our responses has been attached.

R. Alex Thompson

(On behalf of the authors)

Summary of our revisions:

Throughout these comments, three main concerns were brought up by the reviewers. First, our emphasis on starch dynamics as a proxy for total NSC was viewed as potentially problematic. We previously analyzed total NSC, as well as starch and sugar individually, and prior results suggested these dynamics were primarily driven by a depletion of starch. Our updated manuscript now includes an analysis of total NSC in addition to sugar and starch.

The second point of concern focused on our use of relative values (% of maximum) for the traits we measured in this study. There are several reasons for this approach, and we emphasize our reasons for using it in the updated manuscript (see below for specific line-by-line updates). Growth, NSC, and photosynthesis were measured at the individual level, but analyzed at the species-level. To use the data from all trees for a species in a single analysis when trees had different rates of growth and photosynthesis, we calculated relative values to re-scale the raw data and preserve variance.

Finally, there was a concern about the use of a single maximum value for scaling measurements. Although expressing our data as a % of maximum should have no effect on the general trends of our analysis, we include an additional document with our manuscript titled “Sensitivity Analysis” where we test the effect of our choice of maximum value on the results. After re-evaluating all our data with shifted maximum values: +10%, -10%, +25%, and -25% of the observed maximum value. Changing the maximum value had no effect on our findings, thus our sensitivity analysis highlights the robustness of our results, thus we feel confident in our approach. We address specific concerns below.

REVIEWER COMMENTS

Reviewer #1 (Remarks to the Author):

Thompson et al. put together a very impressive conglomeration of long-term datasets measuring water potential, radial growth, photosynthesis and carbohydrate storage. Data are collected in the American Southwest and are from a combination of *in situ* measurements and experimental drought treatments. Using their data, they explore the long-debated source versus sink limitation in storage formation, with respect to drought. They ask the questions:

1. Does growth stop before photosynthesis in response to declining water potentials?
2. Does carbon storage increase as growth declines with more negative water potentials?

Both of these suppose sink limitations, in which growth is more sensitive to environmental stress than photosynthesis, thus carbon continues to be assimilated and added to storage, rather than being put toward growth when under stress. Interestingly, the study found that (1) photosynthesis ceased before growth did in both pine and juniper, suggesting the foundation of the sink limitation hypothesis may not actually be sound. Next, they found that (2) starch stores decreased steadily in response to drought, rather than increase as they predicted.

While I find the first finding very interesting and backed by an impressive dataset, the majority of the paper is focused on the second finding and the question of storage, which I have several major issues with. Overall, the dataset is impressive, but the framing of the data and the analysis of the storage data in particular leaves a lot to be desired.

Major Points:

Comment

1. The recurrent refrain and framing of the paper is that starch (which they claim is the storage molecule) increases following drought and this idea is well known and pervasive in the field (L35-36, L56-57, L177-178). The authors then set out to disprove this with their dataset. I have a two-part issue with this idea:

a. Starch as the only storage molecule of interest I think is inappropriate here. While starch is thought of as long-term storage because it is insoluble, both sugars and starches (combined as NSC) are being put away by plants for future use. Whether it is stored as sugar or starch varies on daily to annual timescales – sugar is regularly converted back and forth into and out of starch daily (Gersony et al., 2020; Tixier, Orozco, Roxas, Earles, & Zwieniecki, 2018), seasonally (Martínez-Vilalta et al., 2016), and in response to environment (Thalman & Santelia, 2017).

b. This flexibility of plants to convert stored NSC back and forth is very well established, particularly in response to drought. Many studies have demonstrated that following drought, starch is broken down into sugar for a number of regulatory functions (eg. Thalmann & Santelia, 2017). Thus, I'm not sure how you come up with Fig 1A, I think your findings illustrated in Fig 1B & C are what I would have predicted for starch given the literature. Indeed, none of the papers you cite to support your assertion that starch accumulates following drought actually say this. Körner et al. 2003 only presents total NSC data and while he shows that total NSC increases following drought, if you return to the original dataset, you find that starch is broken down. Similarly, Wiley and Helliker 2012 only refer to total NSC, starch is in fact never mentioned in the paper, and Monson et al. 2021 is a review of growth-defense tradeoffs, where storage is really just peripheral to the paper.

Thank you for your insightful comment. Your point is well taken and we agree that our explanation for only using starch was not as clear or prominent as it should have been given the complexity associated with NSC dynamics. We have updated our language to refer to total NSCs in the abstract, introductory paragraphs and throughout the updated manuscript (e.g. lines 149,160, Fig. 1) Where useful to improve clarity and discuss results we also refer to sugar and starch in addition to total NSCs (e.g. lines 149, 152, 154). The analysis shown in Figure 1 has been updated to show total NSC instead of starch, and Figures 3 and 4 continue to show both sugar and starch responses separately. Notably, our results highlight that the change in total NSC dynamics is due to the response of starch concentrations, and we show this in Figures 3 and 4.. Now this point, that starch contributes more to the NSC dynamics than sugar, is explained and emphasized in the discussion of the results and in figure captions, rather than assumed early in the paper. We believe this improves the flow of the manuscript and appreciate the comment. Additionally, we have included citations to Gersony et al. 2020 and Tixier et al. 2018 in our methods section (specifically, line 317).

Comment

2. Relatedly, only examining starch and sugar separately and as percentages of their maximum makes interpretation very difficult. While it makes comparisons to other measurements easier, it also may be hiding some significant patterns. For one, it makes starch and sugar look like they are equitable in the amount of energy they may provide. However, whether starch is 2% or 80% of total NSC would drastically alter how one views your findings. Storage should really be

represented here by total NSC, or that metric should at least be included in your analysis to truly understand what is happening with storage post drought.

Thank you. This is an excellent point and we have now updated our analysis shown in Figure 1 with total NSC. Total NSC is the sum of sugar and starch concentrations of each tissue and each tree before considering as percentages relative to maximum total NSC.

Comment

3. Statistics could be much better reported throughout the text, figures, and tables. I have more minor comments below that indicate where it is a particular issue, but an eye throughout to incorporating the results of statistical tests to back up your findings in text and in figures is needed. In general, it would be good to report slope estimate, sample size, and p-value in text to back up your statements. Tables should also be more standardized in their reporting - in some cases like table S2 all relevant information is included, but then in table S6 parameter estimates are not provided.

We thank the reviewer for their considering the reporting of statistics. We address specific concerns below, but update our reporting throughout the text to reflect the suggested changes (e.g. lines 121, 151-152). Tables have been added to report additional estimates of intercepts, slopes, standard errors, and sample sizes where applicable (e.g. Table S8).

Minor Points:

1. The writing would benefit if the terms source/sink limitation were defined upfront (before L86) as used more consistently throughout, as that is what they are ultimately testing.

Sink and source limitation are now introduced at lines 51 and 54, respectively.

2. L142 – provide slope, give N and p-val as well.

Relevant statistics have been updated and are provided in the manuscript (line 152) and Table S7. Specifically, this line has been updated with the following information: “(Figs. 3C, 3D; Slope = -1.14, -1.032⁻⁴²; p <0.05, p = 0.48; n =32 , n = 34, respectively)”

3. L94 – growth did not stop before photosynthesis in both species, or just juniper? Confused because you later say “in contrast”, suggesting species are doing different things.

We removed the phrase “in contrast” to clarify our point here (line 98)

4. L100 – I think a little more detail on the two species and why they are expected to respond to drought differently would go a long way to understanding the results.

*A brief discussion explaining why *J. monosperma* and *P. edulis* respond differently to drought is included on lines 101-111. We state: “The distinct co-limitation patterns of each species reflect their differing tolerances to drought stress. *P. edulis* is considered isohydric compared to the relatively anisohydric *J. monosperma*²². Though many factors influence a species’ position on the an/isohydry spectrum, two dominating factors are the regulation of leaf water loss²⁴ and hydraulic vulnerability²². The relatively early stomatal closure of *P. edulis* is an embolism-avoidance mechanism, preventing water loss through transpiration, avoiding hydraulic failure. *J. monosperma*, owing to its less vulnerable xylem, leverages osmotic adjustment to keep stomata open and potentially maintain a positive growth rate under progressively lower Ψ_{pd} . While an/isohydry may explain the quantitative differences we observed (Fig. 1), we do not test this further. Instead, we focus on questions relating to the qualitative dynamics (i.e. we look within species or along common axes of variation).”*

5. L 224 – are they planted in pots or in the ground?

These are mature trees growing naturally in the field. This detail is clarified on lines 246-248

6. L 268 – At what date did collection of tissue for NSC end?

This information is now updated on line 312. NSC collection spanned March 14th 2012 – October 13th 2016.

7. Fig 1. You show the result of three separate regressions as equations, but report 1 p-value. What does this correspond to?

Updated Fig. 1 caption to reflect that the P-value is the result of an ANOVA comparing the regression lines of growth and photosynthesis.

8. Fig. 2A – While they appear uncorrelated, you should back this up with statistics.

The results of running a linear model on the data in Figure 2A reveal a significant positive correlation between % of maximum growth and % of maximum photosynthesis for J. monosperma only, but with a very low R^2 of 0.05. While statistically significant, photosynthesis only explains 5% of growth, a very low effect. This has been updated in the figure caption and discussed further on lines 119 – 122. We state: “Growth and photosynthesis were significantly correlated for J. monosperma yet very little of the variance could be explained ($R^2 = 0.05$). This suggests a latent, underlying driver of this variation.”

9. Fig. 2B – would be helpful to highlight the growing season on graphs for these species and show that there is break in measurements between years.

We have updated Fig. 2B to highlight the break in measurements between years. No growth measurements outside of the growing season were made, and all photosynthesis measurements shown in 2B are truncated to only within the growing season.

10. L108-110; Fig 2C – I am a bit confused, you make the argument in 2A that photosynthesis and growth are uncorrelated, but then they look remarkably coordinated in 2C. I’m not sure entirely how you get from 2C complete dataset to 2A. How are you summarizing data here? It says tree-level observations, but then are you averaging over all months too?

Although there were periods of coordinated growth and photosynthesis (shown in Fig. 2B and 2C), there were also periods where growth could remain high even while photosynthesis was low (Fig. 2A, 2B). Figure 2A clearly shows that the growth and photosynthesis correlation is weak (for J. monosperma) to non-existent (for P. edulis). Figure 2C does show that one should take care in assessing the strength of a correlation by eye from two sets of time series data overlaid on the same figure. We clarify these dynamics by describing the coordinated, yet independent response of growth and photosynthesis to declining water potential (line 175; Fig 4). Figure 4 also shows the limits of this clarification clearly. Additionally, we address differences in data sources across Fig. 2 as an update to the caption, saying: “In A, points represent tree-level observations of growth and photosynthesis from May – September 2013 and 2014 at SUMO. In B, points represent average growth and photosynthesis (across individuals within species) for each month of the growing season at SUMO, while error bars represent standard error. In C, lines represent individual tree Ψ_{pd} measured monthly through time at MDB (left) and SUMO (right).”

11. I don't have issue that you are drawing on multiple datasets from different individuals, but I think it could be more clear and transparent in the text what comes from where and when.

Possibly a table or timeline graphic could be helpful.

*In cases where data from both experiments is discussed, we have taken care to explicitly mention data sources (see generally lines 134-141. For example, at line 136 we state “2013 at SUMO was the only year that *J. monosperma* ever experienced ψ_{pd} low enough for growth and photosynthesis to cease.”).*

Cited Papers

- Gersony, J. T., Hochberg, U., Rockwell, F. E., Park, M., Gauthier, P. P. G., & Holbrook, N. M. (2020). Leaf Carbon Export and Nonstructural Carbohydrates in Relation to Diurnal Water Dynamics in Mature Oak Trees. *Plant Physiol*, 183(4), 1612-1621. doi:10.1104/pp.20.00426
- Martínez-Vilalta, J., Sala, A., Asensio, D., Galiano, L., Hoch, G., Palacio, S., . . . Lloret, F. (2016). Dynamics of non-structural carbohydrates in terrestrial plants: a global synthesis. *Ecological Monographs*, 86(4), 495-516. doi:10.1002/ecm.1231
- Thalmann, M., & Santelia, D. (2017). Starch as a determinant of plant fitness under abiotic stress. *New Phytol*, 214(3), 943-951. doi:10.1111/nph.14491
- Tixier, A., Orozco, J., Roxas, A. A., Earles, J. M., & Zwieniecki, M. A. (2018). Diurnal Variation in Nonstructural Carbohydrate Storage in Trees: Remobilization and Vertical Mixing. *Plant Physiol*, 178(4), 1602-1613. doi:10.1104/pp.18.00923

Reviewer #2 (Remarks to the Author):

In the manuscript entitled “No carbon storage for growth-limited trees” Thompson et al. used a long term monitoring of predawn water potentials dataset from a juniper-pine woodland (MDB) and a 7 years monitoring of predawn water potentials, radial growth, leaf photosynthesis and NSC concentrations in a rainfall exclusion experiment (SUMO) to evaluate carbon source-sink dynamics under water availability fluctuations. The authors tested the hypotheses that: growth stops before photosynthesis in response to declining predawn water potential (i.e. proxy of water availability in the soil for plants); carbon storage increases as growth declines due to decrease in predawn water potential. The authors conclude that, for the two conifer species analyzed, co-limitation of growth and photosynthesis is rare, except under extreme drought. Different patterns in growth and photosynthesis decline with decreasing predawn water potentials were observed in the two conifer species, in line with

their different drought stress tolerance behaviours. For the same reasons, also starch and sugar patterns with drought progression were different.

The manuscript is well written. I agree with the authors' conclusion that co-limitation of growth and photosynthesis would be the result of a coordinated but independent response to drought. However, under non-drought conditions, it is known that along the growing season several factors determine independently the two processes. E.g. tree phenology operates separately on growth and photosynthesis (e.g., in the end of summer growth decreases or ceases while photosynthesis still could be high compatible with the evergreen habit and to high availability of water due to abundant precipitation). This justifies what observed in fig. 2B in the end of 2013 and 2014 growing seasons. For example, the fact that growth and photosynthesis are uncoupled in the second part/end of the summer (when water availability is high/no drought occurs) allows to increase the storage of NSCs. In addition, photosynthesis and generally leaf gas exchanges depend not only on predawn water potentials but also on VPD. For all these reasons, the lack of relationships between growth and photosynthesis shown in Fig. 2A, given the dataset used (which pools data from the whole growing season), is not surprising.

*Thank you. Figure 2B suggests divergent paths of growth and photosynthesis late in the growing season. While this might indicate a period of "excess photosynthate" and thus increased NSC concentrations, it nonetheless reflects our main hypothesis: that NSC accumulates when growth is low. Though this manuscript specifically considers the circumstance when drought limits growth, the result would be the same. However, Figures 3A and 3B dispute this possibility, showing that when growth is low (i.e. August or September, Figure 2B) NSC still declines (in the case of *P. edulis*) or remains unchanged due to large conversions of starch to sugar (in the case of *J. monosperma*). If the hypothesis that NSC increases when growth declines was supported by our data, further investigation into the specific role of seasonality would be necessary.*

Much of our paper is dedicated to the analysis of growth and photosynthesis interdependencies (or lack thereof). As pointed out by this comment, a correlation between growth and photosynthesis is likely time-dependent. Figure 2A obfuscates this "seasonal correlation" by overlaying growth and photosynthesis from different periods. Yet, if growth and photosynthesis were interdependent processes, this seasonality would not appear. Under interdependence, high growth could be achieved only during periods of high photosynthesis. This is contrary to what we observed (Fig. 2B). Furthermore, we show that growth and photosynthesis are orthogonal (Fig. 4B). If growth depended on photosynthesis, even for a small part of the year, the angle between these vectors would be much less than 90°.

Looking at methods, sampling and data handling for NSC should be made more clear:

1- The authors present an NSC dataset, showing starch and sugars dynamics. However, given that they took samples from different tree organs in each tree individual, it is not clear how the authors used the different NSC samples collected. In particular, it is not clear how the authors could pool leaf and stem NSC in a same dataset to explain NCS variation dynamics. How did they scale them up to the whole tree level? Leaves and stems can have huge differences in NSC concentrations (and in the total NSC contribution for the plant) and also differ widely in terms of NSC dynamics over time (Hoch et al., 2003). This because leaves are photosynthesis sites and are more affected by photosynthesis dynamics. Stem NSCs are dependent on phloem transport, long term storage, growth etc.. Therefore, if not already done, it could be important to perform the analyses separately in leaves and stems (and also roots, and bole) to check for organ-dependent dynamics. Figs. 3a and 3B show a high variability in starch variation (% of maximum) over water potentials: might this be explained by distinct behaviors among organs?

We thank the reviewer for noting the potential for tissue-level variation. On lines 330-333 we clarify our approach for scaling NSC from tissue, to canopy or whole-tree scales. Briefly, we averaged NSC concentrations across all tissues being considered (e.g. the mean of stem and needle for canopy NSC). Since our analyses of NSC, including sugar and starch, depend on the relationship between these and water potential, we have added three additional analyses to assess these relationships for leaves, twig, bole and roots separately. Generally, the results of these were consistent with our overall relationship between whole-plant NSC and component sugar and starch with water potential (Figures S8-S10). Where differences between tissue and whole tree responses were observed, this was due to large decreases in starch concentrations and increases in sugar concentrations. This indicates osmotic adjustments that may offset the tissue-wide observations of starch depletion in both species. These analyses are in our updated supplementary file included in the resubmission (Figs. S8-S10).

2-NSC (especially starch) concentrations usually change a lot during the day (especially in leaves, where starch accumulates during the day and is depleted during the night; but also in stems variations can be large, e.g Tixier et al. 2018, Plant phys.). The authors did not specify if samples were taken in a specific time of the day or not. If samples were taken at any time, this could be a major issue in the use of these data, if they do not demonstrate that daily variation in their trees is negligible.

The temporal dynamics of NSC are important to consider here. To reduce the potential for unwanted variability due to sample timing, all NSC samples were collected between

11:30am and 1:00pm and immediately placed in liquid nitrogen to stop enzymatic activity. We have updated the manuscript accordingly, on line 318.

3-The authors should specify what they mean for stem: was it a branch? Of which age (this can highly change NSC concentration)? Did they keep the bulk stem NSC (bark + wood)? THIS should be specified.

Stems refer to twig samples of approximately 0-5 years in age for P. edulis, and similar recent growth for J. monosperma (although dating of tissue was not possible for this species). Bulk stem samples were processed (phloem and bark), though bark consisted of a minimal portion of the bulk mass due to the relatively young age of the tissue. This has been clarified on line 312-317. Bark was also included in root and bole samples, clarified on line 317.

At last, the authors studied two conifer species. Different patterns could be found in e.g. broadleaf deciduous trees, which differ from conifers in many aspects, like e.g. the amount of stored NSCs in the stem, phenology, leaf habit..

At lines 215-219 we briefly discuss the scope of these results and applicability to other tree species.

Minor comments:

L.98: fig 1 B, not A, right?

Fixed.

L.100: fig. 1C, not B, right?

Fixed.

Ll. 120-122: the authors mention values of Ψ_{predawn} reached in 2013 and 2014, referring to Fig. 2C (displaying growth and photosynthesis), but there are no values of $\Psi_{\text{pre-dawn}}$ in the figure mentioned. It is also not clear to which experiment the authors refer to with those water potential data: SUMO, MDB, both? Please specify.

The reference to predawn was made in error and has been updated to reflect the % of maximum growth and photosynthesis values shown on the y-axis of Fig. 2C. See lines 140-141 for clarification. We now state: “lines are estimates of growth and photosynthesis derived from ψ_{pd} ”

L. 183-184: I agree. Belowground compartments under drought, for example, are understudied in this sense..

L. 262: is 0 RH correct?

0% RH is correct.

Reviewer #3 (Remarks to the Author):

The manuscript by Thompson et al. uses two datasets that include measurements of water potential, growth and photosynthesis to assess whether growth and photosynthesis are decoupled under decreasing plant water potential and whether growth decline led to increased NSC storage. A direct test of the decoupling of growth and photosynthesis is rare in most ecosystems and remains an important question for informing vegetation models, especially as climates continue to change. I found the manuscript well written and valued the assessment of the responses to decreasing water potential. I would suggest readdressing your two questions in the final paragraph and providing a clear answer/synthesis of the conclusion to those two questions. I would qualify that your results may be specific to semi-arid gymnosperms and not necessarily applicable to other systems or functional groups.

To expand and clarify our discussion, we have directly restated our guiding questions on lines 197-200. We state: “In this study we directly addressed two questions relating to the drought response of trees: 1.) Does growth stop before photosynthesis? and 2.) Does NSC accumulate during periods of reduced growth?.”

On lines 215-219, we specifically discuss the limitations of our study as only applying to semi-arid conifers but suggest future research should focus on other species, ecotypes, and functional types as well. We state: “it remains unclear whether the dynamics we observed in this study apply to trees in other ecosystems, with distinct climates, evolutionary histories, or functional types. This study however, provides the first evidence that growth does not stop before photosynthesis, that NSC does not accumulate in all trees during drought, and that growth does not depend on photosynthesis.”

Despite my general positive view of this manuscript, I found the methods missing simple information and the presentation of the frequency, timing and duration of measurements difficult to follow for each separate dataset. Due to that lack of clarity, it is difficult to assess the validity of

the analyses, especially as the response variables are on a relative scale and the calculation of that relative variable can alter the results. Therefore, I strongly suggest the methods are revised with greater detail and clarity (details below).

In response to comments from the other reviewers and the detailed comments below, we have made numerous revisions to our description of methods. In particular, we have restructured the methods section to include a brief introduction that introduces the fusion of two datasets in our analysis. This first paragraph explains to the reader what they should expect throughout the methods section (see lines 232 – 236). Briefly, we have restructured the methods to first introduce data collected at SUMO, and then introducing data collected at MDB. This SUMO-then-MDB structure is maintained throughout the methods section. Where data from MDB is not collected, a short sentence indicating this is provided (e.g. line 333).

Beyond the methods section, we have made several major updates to the manuscript. In the supplementary file, we include updated tables of relevant statistics (see table S8) and tissue-level NSC analyses. Though our initial manuscript focused on starch variation as a proxy for total NSC, we have updated our analysis to focus on total NSC concentrations (Fig. 1) or total NSC and sugar and starch (Fig. 3). This updated approach improves the clarity of our manuscript. We discuss variation in total NSC, sugar and starch at lines 149 – 152.

The analyses we have included in our manuscript are all at the canopy, or whole-tree sales. We have included tissue-specific analyses in Figs S8-S10, clarifying our scaling approach (an average of concentrations) in the methods section at lines 330-333. Finally, we address the potential for different maximum values to affect our results and interpretation in a new supplementary file titled Sensitivity Analysis. In sum, our results are robust to the choice of maximum value. Though the quantitative results change with changing maximum values (i.e. the slope varies with different maximum value choices), the basic dynamics and interrelationships did not change. Growth and photosynthesis were still co-limited with declining water potential, and NSC declined rather than increased as growth decreased.

Detailed points

L95: I believe based on Fig. 1 this value should be 33.3 not 0.33.

This was a typo and has since been fixed at line 95.

L120-122: Typically, averages are given with some metric of variance. Please provide.

We have updated the reporting of average values to include standard error estimates (e.g. line 134, line 136).

L125: What difference? Between the species or years?

This difference refers to a difference among years, not species. We have updated this in the text on line 141.

L135: Why are figures being presented out of order? I would switch A,B with C,D in figure 3 to maintain order.

We have reordered the figures in Fig. 3 to follow this suggestion.

L150: Two Bs, delete one.

Fixed.

Methods

It is unclear to me if growth was measured on the same trees that had measurements of water potential and photosynthesis. This is critical for interpreting the robustness of the data and analyses.

This is an important point. The answer is yes, growth was measured on the same trees that had water potential and photosynthesis measurements. We have updated the methods section to reflect that only trees with measurements for all relevant parameters in an analysis are included in our analyses (line 264-265). On lines 265-266 we now state “all trees assessed in this study at SUMO had measurements of growth, photosynthesis, water potential, and NSC”. Measurements were collected on other trees at the SUMO site than those measured for growth, but those data are outside the scope of this manuscript.

In general, I found the methods difficult to follow due to the fact that there are datasets. Would it be useful to reorder the methods into sites and broadly describe what was measured, for what periods and frequency. Then a detailed section on the methodologies of water potential, photosynthesis, etc.

To ease the flow of reading the methods we have followed this suggestion. First, we introduce the basic structure of the methods section to give the reader an idea of what to expect, when specific experiments are mentioned, and when they are not (lines 232-236). Briefly, we describe both the MDB and SUMO sites together, since they are nearby and have the same site descriptions. For the following sections, we first introduce

measurements made at SUMO and then introduce measurements made at MDB. When measurements were not made at one of the sites, we clearly state this to avoid any confusion of which data comes from which study.

For example,

L251: It is not clear how often measurements were made for SUMO.

We have adjusted the structure of the methods section to clarify this (e.g. lines 274, 296-297, 312).

L215-216: Monitored for what? How often were trees measured for growth in MDB

No growth was measured on trees at MDB, but rather growth and photosynthesis were estimated from the equations in Figure 1. We have clarified this in the methods specifically on lines 267-273, lines 288-289, and line 310. Our changes to the structure of the methods section, in response to your earlier comment, should make this much easier for the reader to follow.

L286-287: It is not clear how disjointed the sampling was at MDB. Simple measurement details have been glossed over that are essential to understanding the analyses, especially as all the results are presented on relative to maximum scales.

*We have clarified the specific details of sampling protocols and any differences between MDB and SUMO in the methods. We emphasize that only water potential measurements were made on MDB trees. MDB water potential measurements were made monthly, as is now stated clearly on line 296-300. We state: “At MDB, xylem water potential was measured every month between 1992 – 2016. Measurements were made on 5-6 individuals of *J. monosperma* from 1992 – 2012, and 11-14 individuals of *J. monosperma* from 2013 – 2016. Between 1992 – 2016, water potential measurements were made on 5-7 individuals of *P. edulis*.” This should be clearer following the restructuring of the methods suggested in an earlier comment.*

L289-296: So, a single measurement was used as the maximum for an individual. Would not an average of the top 95% or something similar be more conservative or even the average maximum of all individuals? Please provide more support for the selection of a single value as the maximum or test the effects of alternatives calculations on the results.

The choice of a value against which all other observations were set relative to was largely arbitrary and served the function of allowing us to compare data from separate individuals at the species level. However, we thank the reviewer for this important observation. To address this, we conducted a sensitivity analysis (see supplementary file Sensitivity Analysis)

and tested the choice of maximum value on our results. Briefly, we varied the maximum value by +10%, -10%, +25%, and -25% and re-ran all analyses. Our results suggest no difference in the ultimate dynamics of our system and that our results are robust to the choice of maximum value. We now discuss this sensitivity analysis in our manuscript on lines 417-424 and direct readers to the supplementary file Sensitivity Analysis on (line 419).

L306: How were % maximum NSC determined?

We have updated how % of maximum NSC was determined on lines 377-381. We state: “we determined the tree-level maximum NSC, sugar, and starch, and expressed all subsequent values as relative to that”

L306-307: Pooling across all treatments? Be more explicit.

*This has been clarified on lines 360-362. We now state: “we chose to evaluate differences in thresholds and NSC dynamics on all trees at the species-level, increasing our sample size from ~3 trees per species per treatment to 11 individuals of *P. edulis* and 12 individuals of *J. monosperma*”*

L339-341: PCA depends on normally distributed data. Did the data meet this requirement? Were the relative values also used in the PCA?

The assumptions of PCA are minimal. PCA works by taking linear combinations of random variables and expressing them as an eigenvalue, a metric of the norm or stretch of all of the variables in some orthogonal direction. Thus, PCA makes no assumption of data normality (see Hastie et al. 2009 for a detailed proof). However, we did scale and center the data prior to analysis to ensure each vector had unit variance and the scale of one variable did not overwhelm the results. All analyses were done the relativized data, including PCA. This is stated on lines 409-410: “we were able to analyze the centered and scaled data using a correlation matrix of the data”.

References

Hastie, Trevor, et al. *The elements of statistical learning: data mining, inference, and prediction*. Vol. 2. New York: springer, 2009.

REVIEWER COMMENTS

Reviewer #1 (Remarks to the Author):

I greatly appreciate the authors efforts to address all comments and the clarity of their methodology has vastly improved. It has made your experimental design and plots much more clear to me what they are representing. However, as a result I think the inclusion of the MDB data in this study is not appropriate. Given only water potential was measured at MDB, you extrapolated what growth and photosynthesis would be based on the relationships measured in SUMO. While significant, the relationship between growth and water potential only explain a small fraction of the variation in growth ($R^2 \sim 0.1$). You then examine these extrapolated measurements for patterns in their variation (how often does co-limitation occur). The problem here is by correlating both growth (variable A) and photosynthesis (variable B) to the water potential (variable C), you are unintentionally creating artificial relationships. If you regress A to C and B to C, then examine the predicted A to predicted B, you will always find a relationship, even if A and B are uncorrelated initially. While maybe this would give you some information on trends, you can't really go through and examine how many times a non-average value (ie co-limitation) occurs with much confidence. The inclusion of the MDB could be a line in the paper that supports your argument on a longer time-scale with many caveats, but is certainly not convincing as a hypothesis test or selling point of results. Indeed, it only really gets one line of explanation in the paper (L123-125).

Alternatively, the SUMO data are certainly interesting to a general plant physiology audience and help explain what might be going on in these two desert species under drought. However, its one site and two species, so while an interesting piece of evidence, I do not think it conclusively disproves or proves source/sink limitation hypotheses.

Reviewer #2 (Remarks to the Author):

I have reviewed the manuscript in the previous review round. Overall, the manuscript has improved and proper answers have been given to the comments/requests. However, based on the authors' clarifications given in this reviewed version, I see a major issue on NSC analyses and authors' related discussion, that should be fixed. Based on my previous request (Reviewer 2) to specify how NSCs were calculated (samples came from 4 different organs, i.e. leaves, roots, twigs and bole), the authors indicated that they averaged the leaf, bole, twig and root values for each sampling time and individual. This approach of averaging NSC concentrations among organs, without weighting them

based on organ biomass (and, therefore, NSC) contribution, is misleading and uncorrect. In terms of biomass, roots and bole should likely count more than e.g. leaves. Moreover, in this reviewed version of the manuscript, the authors reported (as suggested by me in the previous round) the single NSC-water potential dynamics divided by organ. The result is that NSC (total, as well as sugars and starch separately) varied differently when considering the single organs with respect than averaging them equally (compare e.g. fig. 1, 3 with fig. S8, S9, S10). This approach of averaging NSCs does not consider the physiological meaning behind all these data.

As data on relative biomass contribution of the different organs is likely not available (if yes, it would be great to do this weighting, but I understand this data on the mature trees may be difficult to obtain), the authors should in my opinion comment NSC variation over water potential on an organ-based manner. The overall message given by the authors does not change (i.e. NSC do not always accumulate during drought), but this distinction is physiologically meaningful and should be always carefully taken into account by modellers.

Here below specific comments based on the authors' text.

At line 149: Fig. 1B and C, show an average NSC concentration among organs: this approach of averaging organs does not consider that each organ weights differently in terms of total NSC storage (indeed, bole and roots NSC stocks likely weight more as they should usually count for more biomass than e.g. leaves, therefore storing a lot more NSCs than e.g. leaves). In any case, the "decrease" in NSCs (averaged among organs) with decreasing water potential is weak. Moreover, Fig. S8 (which shows NSC separately in leaves, bole, roots and twigs) does not support the sentence at l. 149 for *J. monosperma*, for which there is barely no decline in NSC concentration (average NSC is the same over water potential for all organs, or it does even slightly increase in roots with decreasing water potentials).

ll. 162-167: in line with the previous comments, here commenting the "average" dynamics (not weighted by organ contribution to overall NSC pools), is misleading. As stated by the authors, the "average" NSCs dynamics say that sugar concentrations were kept unchanged for *P. edulis*, and this is true. However, this is driven by apical organs NSC dynamics (leaves and young twigs). On the contrary, bole and roots showed an increase in sugars with decreased water potentials (Fig. S10). I think this is important to be underlined, as an actual sugar accumulation seems to occur in the main storage organs (roots, bole), while perhaps the osmotic potential in leaves and young twigs was sufficient to keep these organs alive at the reached water potentials (much higher than those reached by *Juniperus*, indeed).. Please comment this point.

Reviewer #3 (Remarks to the Author):

I was largely positive on my first review of this manuscript but had raised questions regarding the use of relative values in the methods. The authors have made a great effort to address the lack of clarity in the methods section that I raised earlier. I especially appreciate the sensitivity analyses and clarification of the measurements made at the SUMO site. I feel that my comments have been fully addressed and I have no further issues with the manuscript.

Dear Reviewers,

We thank you for your thoughtful and helpful comments. We have adjusted the manuscript as requested and respond to each specific comment below. All of our responses are *italicized* and a fully formatted document of our responses has been attached.

R. Alex Thompson
(On behalf of the authors)

Summary of our revisions:

Two primary concerns were brought to our attention by Reviewers 1 and 2. The first issue was the extrapolation of data from SUMO to MDB. Our approach created a spurious correlation between growth and photosynthesis. The result was that the predictions shown in the MDB data could be unreliable. Due to this issue, we have removed the MDB data from the main figures and placed it in the supplementary. We have updated the text of the manuscript to reflect the tentativeness of this result, and instead choose to highlight a need for further research on this topic. The second concern was related to our averaging of NSC data across tissues. As pointed out by Reviewer 2, this is inherently problematic due to unequal contributions of each tissue. To account for this, we expand our discussion of the data to include the tissue-level responses. We also include new figures that reflect these tissue-level dynamics. Finally, the reviewers raised several concerns related to the generality of our results. We have made several changes to the manuscript (including the title) that highlight both the limited scope of these results (two species at one site) and the need for studying this topic more broadly. We italicize our response to specific comments below.

REVIEWER COMMENTS

Reviewer #1 (Remarks to the Author):

I greatly appreciate the authors efforts to address all comments and the clarity of their methodology has vastly improved. It has made your experimental design and plots much more clear to me what they are representing. However, as a result I think the inclusion of the MDB data in this study is not appropriate. Given only water potential was measured at MDB, you extrapolated what growth and photosynthesis would be based on the relationships measured in SUMO. While significant, the relationship between growth and water potential only explain a small fraction of the variation in growth ($R^2 \sim 0.1$). You then examine these extrapolated measurements for patterns in their variation (how often does co-limitation occur). The problem here is by correlating both growth (variable A) and photosynthesis (variable B) to the water potential (variable C), you are unintentionally creating artificial relationships. If you regress A to C and B to C, then examine the predicted A to predicted B, you will always find a relationship, even if A and B are uncorrelated initially. While maybe this would give you some information on trends, you can't really go through and examine how many times a non-average value (ie co-limitation) occurs with much confidence. The inclusion of the MDB could be a line in the paper that supports your argument on a longer time-scale with many caveats, but is certainly not convincing as a hypothesis test or selling point of results. Indeed, it only really gets one line of explanation in the paper (L123-125).

Alternatively, the SUMO data are certainly interesting to a general plant physiology audience and help explain what might be going on in these two desert species under drought. However, its one site and two species, so while an interesting piece of evidence, I do not think it conclusively disproves or proves source/sink limitation hypotheses.

Thank you for pointing out these issues. We agree that our extrapolation for the MDB data may create a spurious correlation, and thus have removed this analysis from the main paper. A separate MDB figure has been placed in the supplementary (see Fig. S11).

- 1. L108: We have removed the reference to using MDB data to test whether growth was limited more frequently than photosynthesis*
- 2. updated the text to reflect Fig. S11 (L163) and that the data shown there is extrapolated from water potential data*
- 3. We have also added a brief discussion of the limitations of our extrapolation approach from L280 – 284, where we state: “The extrapolated data from ψ_{pd} to growth and*

photosynthesis suggests that drought conditions severe enough to drive this sink-source co-limitation may be rare in this ecosystem Fig. 2; Fig. S11). While this suggests that carbon storage may have increased more often than it decreased, direct and simultaneous measurements of ψ_{pd} , growth, photosynthesis, and NSC are needed.”

We also thank the reviewer for pointing out the limitations of our study. To clarify that these results only apply to two species at one site, we have made several changes:

1. *We started by updating the title to “No Carbon Storage for Growth-Limited Trees in a Semi-Arid Woodland”*
2. *L131-133: changed “Can the co-limitation of growth and photosynthesis under progressively lower ψ_{pd} be explained as the control of growth by photosynthesis (or vice versa)?” to “Can the co-limitation of growth and photosynthesis under progressively lower ψ_{pd} be explained as the control of growth by photosynthesis (or vice versa) in these two species?”*
3. *L255-256: changed “To further quantify the relationship between growth and photosynthesis” to “To further quantify the relationship between growth and photosynthesis in these two species”*
4. *L274-275: changed “In this study we directly addressed two questions relating to the drought response of trees:” to “In this study we used two co-occurring conifer species growing in a semi-arid ecosystem to address two questions relating to the drought response of trees:”*
5. *L278-280: changed “We demonstrated that growth and photosynthesis decline concomitantly in response to drought.” to “We demonstrated that growth and photosynthesis decline concomitantly in response to drought in both *J. monosperma* and *P. edulis*.”*
6. *L294-296: To clarify that our results are limited to the species we studied, we added: “Our results suggest that NSC does not increase during drought in all species. It remains unclear whether the dynamics we observed in this study apply to trees in other ecosystems, with distinct climates, evolutionary histories, or functional types.”*
7. *L311-312: To the same paragraph as change #6 above, we also added: “Future work should aim to test these observations in other species in a variety of environments.”*

Reviewer #2 (Remarks to the Author):

I have reviewed the manuscript in the previous review round. Overall, the manuscript has improved and proper answers have been given to the comments/requests. However, based on the authors' clarifications given in this reviewed version, I see a major issue on NSC analyses and authors' related discussion, that should be fixed. Based on my previous request (Reviewer 2) to specify how NSCs were calculated (samples came from 4 different organs, i.e. leaves, roots, twigs and bole), the authors indicated that they averaged the leaf, bole, twig and root values for each sampling time and individual. This approach of averaging NSC concentrations among organs, without weighting them based on organ biomass (and, therefore, NSC) contribution, is misleading and incorrect. In terms of biomass, roots and bole should likely count more than e.g. leaves. Moreover, in this reviewed version of the manuscript, the authors reported (as suggested by me in the previous round) the single NSC-water potential dynamics divided by organ. The result is that NSC (total, as well as sugars and starch separately) varied differently when considering the single organs with respect than averaging them equally (compare e.g. fig. 1, 3 with fig. S8, S9, S10). This approach of averaging NSCs does not consider the physiological meaning behind all these data.

As data on relative biomass contribution of the different organs is likely not available (if yes, it would be great to do this weighting, but I understand this data on the mature trees may be difficult to obtain), the authors should in my opinion comment NSC variation over water potential on an organ-based manner. The overall message given by the authors does not change (i.e. NSC do not always accumulate during drought), but this distinction is physiologically meaningful and should be always carefully taken into account by modellers.

Thank you for emphasizing these issues with our analysis of the unweighted average NSC, sugar, and starch across all tissues. The intent was to show the average response of each tissue to declining water potential though your points that this obfuscates tissue-specific dynamics and makes assumptions about the relative differences among organs are well-taken. We have updated the manuscript to emphasize tissue-specific variation as well as additional comments on the physiological meaning of these dynamics. Our specific changes are:

L173-208: "On average, NSC in the leaves and twigs of both species decreased with decreasing ψ_{pd} (Fig. 3; slope = -1.2 for *J. monosperma*, -9.4 for *P. edulis*). Similar trends were

observed for the bole and roots of *P. edulis*. In contrast, overall NSC in the roots of *J. monosperma* had little change with water stress (slope = 0.013, p=0.3.). However, roots saw a relatively strong increase in sugar concentrations (slope =0.1, p<0.05) and decline in starch (slope = -0.05, p<0.05; Fig. 3C). To support increased water acquisition from the soil and maintenance of cell turgor in the canopy under low ψ_{pd} our data supports starch conversion to sugar for osmotic purposes in this species²⁴ (Fig. 3C). Since *J. monosperma* maintained photosynthesis under relatively low ψ_{pd} (Fig. 1), it is possible that sugars were transported from the canopy to roots, yet the slight trend in total root NSC does not strongly support this (Fig. 3C). *P. edulis* also exhibited steep increases in bole and root sugar concentrations (slope = 0.19 for the bole and 0.2 for the roots, p<0.05; Fig. 3C) but virtually no change in leaf and twig sugar (slope = -0.007 for leaves, 0.005 for twigs, n.s.). The early onset of source limitation in *P. edulis* (Fig. 1) is known to preserve water storage in canopy tissues, perhaps avoiding the need for starch to sugar conversion in these tissues for osmotic purposes.²⁴ Although *J. monosperma* and *P. edulis* employ distinct physiological responses to water stress, our results indicate a decrease rather than an increase in overall NSC during drought (Fig. 3A,B).

We observed no significant change in NSC or starch with decreasing growth in *J. monosperma* (Fig. 4). Under extremely low growth, canopy sugar concentrations in *J. monosperma* increased logarithmically (slope = 1.14, p<0.05). Mirroring the response to ψ_{pd} , *P. edulis* showed a significant decrease in leaf (slope = -0.54, p<0.05) and twig (slope = -1, p<0.05) starch which may have driven the significant increase in bole sugar concentrations (slope = 0.57, p<0.05) as growth declined. Alternatively, leaf and twig starch in *P. edulis* may have been converted into sugar and used for metabolism, preventing us from detecting a significant increase in sugar concentrations in these tissues. This may explain the significant decrease in leaf (slope =

-0.26, $p < 0.05$) and twig (slope = -0.70, $p < 0.05$) NSC in *P. edulis*. There was nearly a 1:1 conversion of *P. edulis* root starch (slope = -0.6, $p = 0.08$) into sugar (slope = 0.56, $p = 0.06$) leading to no change in root total NSC with decreasing growth (slope = -0.008; $p = 0.9$). These results reinforce our previous observation (see Fig. 3) that NSC does not increase when drought limits the growth of trees in a semi-arid woodland.”

Here below specific comments based on the authors’ text.

At line 149: Fig. 1B and C, show an average NSC concentration among organs: this approach of averaging organs does not consider that each organ weights differently in terms of total NSC storage (indeed, bole and roots NSC stocks likely weight more as they should usually count for more biomass than e.g. leaves, therefore storing a lot more NSCs than e.g. leaves). In any case, the “decrease” in NSCs (averaged among organs) with decreasing water potential is weak. Moreover, Fig. S8 (which shows NSC separately in leaves, bole, roots and twigs) does not support the sentence at l. 149 for *J. monosperma*, for which there is barely no decline in NSC concentration (average NSC is the same over water potential for all organs, or it does even slightly increase in roots with decreasing water potentials).

Thank you for pointing out this difference. To remedy this issue, we have changed Figure 3 to show both the average response and the tissue-specific response to declining water potential. In addition, we have created a new plot of the regression of NSC, sugar, and starch against % of maximum growth (previously Fig. 3A and 3B) that includes tissue-specific responses. This is now Figure 4. Both of these new figures are provided below. In addition, we now introduce the discussion of NSC dynamics by discussing tissue-specific variation (see response above referencing L173-208).

Figure 3. NSC does not increase in *J. monosperma* and *P. edulis* during drought. As ψ_{pd} becomes more negative, both species reallocate carbohydrates from the canopy to the roots (C). In *J. monosperma*, this is likely to increase water supply to the roots and allow stomata to remain open. Additionally, *J. monosperma* increased the sugar concentrations of apical tissues, consistent with osmotic adjustment under drought. In *P. edulis*, osmotic adjustment of apical tissues did not occur as stomata closed relatively early. Instead, increased sugars in the bole and roots likely supported metabolism during extended periods of stomatal closure. These tissue-specific patterns are reflected by the unweighted averages shown in (A) and (B). Points in (A) and (B) are observations of canopy sugar (blue points and line) and starch (yellow points and line). Lines represent the line of best fit, determined using the Akaike Information Criterion (Table S11). Grey areas around lines and on bars indicate 95% confidence intervals. A separate analysis of needle, twig, bole, and root NSC for June only can be viewed in Fig. S3. Tissue-specific regressions can be found in Figs. S8-S10.

Figure 4. NSC does not increase in *J. monosperma* and *P. edulis* as growth declines. As growth becomes limited, NSC, starch, and sugar concentrations do not significantly change in *J. monosperma* (C). In *P. edulis*, a significant decline in NSC, driven by a decline in starch, may drive the significant increase in sugar concentrations in the bole. These tissue-specific patterns are reflected by the unweighted averages shown in (A) and (B). Points in (A) and (B) are observations of canopy sugar (blue points and line) and starch (yellow points and line). Lines represent the line of best fit, determined using the Akaike Information Criterion (Table S11). Grey areas around lines and on bars indicate 95% confidence intervals. Tissue-specific regressions can be found in Figs. S12-S13.

L1. 162-167: in line with the previous comments, here commenting the “average” dynamics (not weighted by organ contribution to overall NSC pools), is misleading. As stated by the authors, the “average” NSCs dynamics say that sugar concentrations were kept unchanged for *P. edulis*, and this is true. However, this is driven by apical organs NSC dynamics (leaves and young

twigs). On the contrary, bole and roots showed an increase in sugars with decreased water potentials (Fig. S10). I think this is important to be underlined, as an actual sugar accumulation seems to occur in the main storage organs (roots, bole), while perhaps the osmotic potential in leaves and young twigs was sufficient to keep these organs alive at the reached water potentials (much higher than those reached by *Juniperus*, indeed).. Please comment this point.

This is an important point and we thank the reviewer for mentioning this. We have updated the discussion of NSC dynamics to interpret this variation. Specifically, we state:

*L175-180: “In contrast, overall NSC in the roots of *J. monosperma* had little change with water stress (slope = 0.013, $p=0.3$). However, roots saw a relatively strong increase in sugar concentrations (slope =0.1, $p<0.05$) and decline in starch (slope = -0.05, $p<0.05$; Fig. 3C). To support increased water acquisition from the soil and maintenance of cell turgor in the canopy under low ψ_{pd} our data supports starch conversion to sugar for osmotic purposes in this species²⁴ (Fig. 3C).”*

*L187-194: “*P. edulis* also exhibited steep increases in bole and root sugar concentrations (slope = 0.19 for the bole and 0.2 for the roots, $p<0.05$; Fig. 3C) but virtually no change in leaf and twig sugar (slope = -0.007 for leaves, 0.005 for twigs, n.s.). The early onset of source limitation in *P. edulis* (Fig. 1) is known to preserve water storage in canopy tissues, perhaps avoiding the need for starch to sugar conversion in these tissues for osmotic purposes.²⁴ Although *J. monosperma* and *P. edulis* employ distinct physiological responses to water stress, our results indicate a decrease rather than an increase in overall NSC during drought (Fig. 3A,B).”*

*L200-208: “Alternatively, leaf and twig starch in *P. edulis* may have been converted into sugar and used for metabolism, preventing us from detecting a significant increase in sugar concentrations in these tissues. This may explain the significant decrease in leaf (slope = -0.26, $p<0.05$) and twig (slope = -0.70, $p<0.05$) NSC in *P. edulis*. There was nearly a 1:1 conversion of *P. edulis* root starch (slope = -0.6, $p=0.08$) into sugar (slope = 0.56, $p=0.06$) leading to no change in root total NSC with decreasing growth (slope = -0.008; $p = 0.9$). These results reinforce our previous observation (see Fig. 3) that NSC does not increase when drought limits the growth of trees in a semi-arid woodland.”*

Reviewer #3 (Remarks to the Author):

I was largely positive on my first review of this manuscript but had raised questions regarding the use of relative values in the methods. The authors have made a great effort to address the lack of clarity in the methods section that I raised earlier. I especially appreciate the sensitivity analyses and clarification of the measurements made at the SUMO site. I feel that my comments have been fully addressed and I have no further issues with the manuscript.

Thank you for your kind remarks. Your suggestions have greatly improved the manuscript.

REVIEWERS' COMMENTS

Reviewer #1 (Remarks to the Author):

I thank the authors for responding to all comments and amending the article accordingly. I only have one minor suggestion - to remove L44-45 in the abstract. You have rightly hedged the statement, but as you can't reconstruct extreme values at all, there's not much you can say with these data (the 25 year record) regarding the presence of absence of co-limitation. Otherwise, it is a nice paper that shows an unexpected lack of NSC buildup under drought.

Reviewer #2 (Remarks to the Author):

The manuscript in this submitted version has clearly improved.

The authors have properly addressed the requests of NSC analyses, considering organ-specific variation in sugars, starch, NSCs, and providing figures that show these distinctions among organs. They have also properly commented these patterns in the discussion.

I don't have any other comment/request.

Dear Reviewers,

We thank you for your thoughtful and helpful comments. We have adjusted the manuscript as requested and respond to each specific comment below. All of our responses are *italicized* and a fully formatted document of our responses has been attached.

R. Alex Thompson
(On behalf of the authors)

Reviewer #1 (Remarks to the Author):

I thank the authors for responding to all comments and amending the article accordingly. I only have one minor suggestion - to remove L44-45 in the abstract. You have rightly hedged the statement, but as you can't reconstruct extreme values at all, there's not much you can say with these data (the 25 year record) regarding the presence of absence of co-limitation. Otherwise, it is a nice paper that shows an unexpected lack of NSC buildup under drought.

*Thank you for kind remarks. As requested, we have removed L44-45 from the abstract.
The complete, updated abstract now reads:*

*“Plant survival depends on a balance between carbon supply and demand. When carbon supply becomes limited, plants buffer demand by using stored carbohydrates (sugar and starch). During drought, NSCs may accumulate if growth stops before photosynthesis. This expectation is pervasive, yet few studies have combined simultaneous measurements of drought, photosynthesis, growth, and carbon storage to test this. Using a field experiment with mature trees, we show that growth and photosynthesis slow in parallel as ψ_{pd} declines, preventing carbon storage in two semi-arid conifers (*J. monosperma* and *P. edulis*). During experimental drought, growth and photosynthesis were frequently co-limited. Our results offer an alternative perspective on how plants use carbon that views growth and photosynthesis as independent processes both regulated by water availability.”*

Reviewer #2 (Remarks to the Author):

The manuscript in this submitted version has clearly improved.

The authors have properly addressed the requests of NSC analyses, considering organ-specific variation in sugars, starch, NSCs, and providing figures that show these distinctions among organs. They have also properly commented these patterns in the discussion.

I don't have any other comment/request.

Thank you for your comments throughout the review process. We feel that including organ-specific variation in NSCs has greatly enhanced the utility of our manuscript and interpretability of our results.